# Thermal conductivity of firn at Lomonosovfonna, Svalbard, derived from subsurface temperature measurements

Sergey Marchenko[1], Gong Cheng[2], Per Lötstedt[2], Veijo Pohjola[1], Rickard Pettersson[1], Ward van Pelt[1], and Carleen Reijmer[3]

[1]Uppsala University, Department of Earth Sciences
[2]Uppsala University, Department of Information Technology
[3]Utrecht University, Institute for Marine and Atmospheric Research

**Correspondence:** Sergey Marchenko (sergey.marchenko@geo.uu.se)

**Abstract.** Accurate description of snow and firn processes is necessary for estimating the fraction of glacier surface melt that contributes to runoff. Most processes in snow and firn are to a great extent controlled by the temperature therein and in the absence of liquid water, the temperature evolution is dominated by the conductive heat exchange. The latter is controlled by the effective thermal conductivity $k$. Here we reconstruct the effective thermal conductivity of firn at Lomonosovfonna, Svalbard, using an optimization routine minimizing the misfit between simulated and measured subsurface temperatures and densities. The optimized $k_*$ values lie in the range from 0.2 to 1.6 $\mathrm{W\,(m\,K)^{-1}}$ increase downwards and over time. The results are supported by uncertainty quantification experiments, according to which $k_*$ is most sensitive to systematic errors in empirical temperature values and their estimated depths, particularly in the lower part of the vertical profile. Compared to commonly used density-based parameterizations our $k$ values are consistently larger, suggesting a faster conductive heat exchange in firn.

## 1  Introduction

Glaciers and ice sheets are important indicators of past and ongoing climate changes. Under the influence of temperature fluctuations at the surface the subsurface glacier temperature also changes. As a basic physical property of a medium, temperature of snow, firn and ice controls multiple processes occurring therein and at the glacier surface.

Climate-induced glacier mass change is strongly affected by the state of snow and firn, where liquid water generated at the surface of glaciers during the ablation period can be refrozen, by this reducing runoff. The magnitude of liquid water refreezing is largely dependent on the subsurface temperature (e.g., Trabant and Mayo, 1985). A snow/firn pack reaching lower temperatures during the winter season is able to refreeze a larger amount of water during and after the ablation season. Warmer snow and firn experience faster metamorphism (e.g., Jordan et al., 2008) and gravitational densification (Ligtenberg et al., 2011). The distribution of temperature under the glacier surface also defines the ground heat flux, which contributes to the energy balance at the surface, and is thus important for simulation of the surface energy fluxes and melt rates. In addition, cold ice is more viscous and less prone to deformation (e.g. Weertman, 1983). Therefore, a cold glacier can be expected to exhibit lower flow velocities, provided that other environmental parameters are equal. Thus the processes of mass and energy exchange

occurring at glaciers are in tight interaction with the subsurface temperature, which needs to be either measured or simulated, if measurements are not possible.

In the absence of liquid water the changes in subsurface temperature are defined by the process of thermal conduction described by Fourier's law (Cuffey and Paterson, 2010, page 403), according to which the heat flux ($Q$) is proportional to its thermal conductivity ($k$) and to the spatial temperature gradient ($\nabla T$): $Q = -k\nabla T$. Since most temperature fluctuations occur at the surface, the dominant direction of heat flux in near surface glacier layers is vertical: $Q = -k\,\frac{dT}{dz}$, where $z$ is the vertical coordinate. Sturm et al. (1997) indicated that several processes contribute to the temperature changes occurring in a subfreezing porous snow and firn, namely: conduction through the rigid ice matrix, conduction through the air in pores and latent heat transport through the pores due to sublimation and condensation of water vapor. To underline that fact and because all three processes are essentially driven by the temperature gradient (e.g., Bartelt and Lehning, 2002) here we use the term effective thermal conductivity ($k$) following Sturm et al. (1997) to describe the ability of snow, firn and ice to transport thermal energy. Along with density ($\rho$) and specific heat capacity ($C$) $k$ is used to calculate thermal diffusivity ($\kappa$) as $\kappa = k\,(\rho\,C)^{-1}$.

Due to the direct connection between $k$ of a medium and temperature changes therein, most empirical estimates of $k$ are based on temperature measurements (Sturm et al., 1997). Continuous measurements of natural temperature fluctuations in snow and firn allow to derive $\kappa$ and $k$ values either using Fourier-type analysis or optimization techniques. Diurnal temperature fluctuations penetrate down to *ca* 1 m and the associated phase lag and/or amplitude dampening occurring with depth can be used to reconstruct the effective thermal conductivity of seasonal snow (e.g., Sergienko et al., 2008; Osokin and Sosnovsky, 2014). Annual fluctuations penetrate down to *ca* 10 m and thus $k$ estimates for thick firn packs using the method require a long data series undisturbed by the influence of liquid water (Dalrymple et al., 1966; Weller and Schwerdtfeger, 1971; Giese and Hawley, 2015). Alternatively, $\kappa$ and $k$ values in the heat equation can be determined by minimizing the misfit between simulated and measured natural evolution of temperature in snow and firn (e.g., Tervola, 1989; Zhang and Osterkamp, 1995; Yang, 1998). Brandt and Warren (1997) applied this method for near-surface snow at the South Pole, Sergienko et al. (2008) for the snow pack on a drifting iceberg and Oldroyd et al. (2013) for the seasonal snow pack on an Alpine glacier.

Another option is to induce a heat flux in the snow pack using a heat source and register the temperature response in either an object with well-known properties that is in contact with both the heat source and the snow or in snow pack itself. The former method is known as the needle probe technique and is widely used to measure in-situ effective thermal conductivity of porous materials including snow (e.g., Lange, 1985; Singh, 1999; Morin et al., 2010). In the latter case the snow sample is placed on a heated plate controlling the vertical heat flux and the respective temperature gradient is measured and the relation between the two values yields $k$ (e.g., Calonne et al., 2011; Riche and Schneebeli, 2013). Sturm et al. (1997) provided an extensive overview of the above named methods and associated uncertainties, which was followed up by further insights into possible biases of needle probe measurements (Riche and Schneebeli, 2010).

A novel technique for estimation of $k$ of snow was suggested by Kaempfer et al. (2005) and further developed by Calonne et al. (2011) and Riche and Schneebeli (2013). The method relies on numerical simulation of processes contributing to the heat flux driven by the temperature gradient based on detailed 3-D X-ray micro-tomography images of the snow matrix. It allows to obtain the $k$ tensor for relatively small snow samples.

Published $k$ values for snow and firn vary from 0.1 to 2.5 W (m K)$^{-1}$ with a strong correlation with density, which justifies the use of density as a proxy to calculate the effective thermal conductivity for modelling purposes (e.g., Sturm et al., 1997; Riche and Schneebeli, 2013). The considerable spread in values suggested by different parameterizations is explained by the inconsistency in applied measurement techniques and associated uncertainties and also by the influence of snow and firn parameters other than density. That can be temperature, grain size and contact area, pore diameter and interconnectivity and anisotropy of the $k$ property.

The purpose of the present study is to reconstruct the values of effective thermal conductivity of a thick snow and firn pack at Lomonosovfonna, Svalbard, based on evolution of subsurface temperature and firn density measured in 2012–2015. Effective thermal conductivity of firn is derived for five distinct periods by minimizing the misfit between the measured and simulated subsurface temperature evolution. The method is promising, particularly for thick firn packs, to our knowledge is has not been applied for the purpose so far, although Sergienko et al. (2008) employed similar routines for a seasonal snow pack and Nicolsky et al. (2007) for permafrost. Fourier analysis applied earlier for a thick firn pack at the Summit of the Greenland ice sheet (Giese and Hawley, 2015) can not be used here due to the influence of melt water and retrieving snow samples for direct measurements using heated plate or needle probe methods is logistically challenging. Estimates of the firn $k$ values are complemented by uncertainty quantification experiments exploring propagation of possible biases in empirical data through the applied models.

## 2 Field data

We use data on subsurface density and temperature evolution collected at Lomonosovfonna, Svalbard, a flat ice field nourishing several outlet glaciers. The field site is at 78.824°N, 17.432°E, 1200 m asl, which is well above the equilibrium line, estimated to be at *ca* 720 m asl (Van Pelt et al., 2012). The local glacier thickness is 192±5.1 m (Pettersson, 2009, unpublished dataset; Van Pelt et al., 2013) of which the firn layer constitutes *ca* 20 m (Wendl, 2014). The accumulation rates estimated from repeated radar surveys (Pälli et al., 2002; Van Pelt et al., 2014) are 0.58–0.75 m w.e. year$^{-1}$. The melt rates simulated by a model describing surface fluxes of energy and mass (Van Pelt et al., 2012) are 0.34 m w.e. year$^{-1}$. The net annual accumulation rates resulting in the relative vertical shifts of the glacier surface are estimated at 1.12 (April 2012–2013), 1.32 (April 2013– 2014) and 0.9 (April 2014–2015) m. The first two values are based on readings at mass balance stake S11 (see Fig. 1a in Marchenko et al., 2017a) and the last one is derived by minimizing the misfit between temperature profiles measured in April 2015 by the thermistor strings installed in April 2014 and in April 2015. The firn pack at Lomonosovfonna is heavily influenced by the percolation and refreezing of melt water (Marchenko et al., 2017b) which results in prominent variability of subsurface stratigraphy at the scale of 10 m (Marchenko et al., 2017a).

## 2.1 Subsurface density

Four shallow firn cores (9.5 cm diameter) were drilled at Lomonosovfonna in April 2012 - 2015 using a Kovacs core drill. The density and stratigraphy profiles (Figure 1) for the first three cores were presented in detail by Marchenko et al. (2017a) along with the details on field and laboratory procedures applied. Similar routines were applied in 2015.

## 2.2 Subsurface temperature

### 2.2.1 Equipment

Subsurface temperature was measured using multiple thermistors grouped in several strings. They were placed in boreholes with 5.5 cm diameter drilled by a Kovacs auger. The boreholes were then backfilled by drill chips and loose snow to minimize the perturbation of the snow and firn media. In 2012–2014 nine thermistor strings were placed in a rectangular $3 \times 3$ grid with a separation of 3 m between neighboring boreholes (Marchenko et al., 2017a, b). In April 2015 only one thermistor string was installed.

All thermistor strings were custom manufactured at Uppsala University using a multi-leaded cable with PVC jacket and uni-curve NTC thermistors. In 2015 the cable was placed in multiple 2 m long rigid plastic tubes to ensure a precise and constant separation between neighboring sensors. The sensors were fed through holes in the tubes and fixed at their outer surface for a better contact with the sounded environment.

The thermistor strings were scanned using a Campbell Scientific CR10X data logger and several relay multiplexors. Each thermistor was connected in series with a reference resistor and precisely measured excitation voltage was applied to the circuit. Voltage drop over the reference resistor was measured and then converted first to corresponding resistances of the sensors and then to temperature values using Ohm's law and recommendations provided by thermistor manufacturer.

The evolution of subsurface temperature was recorded during four time periods: 21 April – 19 October 2012, 22 April – 12 July 2013, 17 April 2014 – 11 April 2015 and 15 April – 9 July 2015. The frequency of measurements is shown by the color bar in bottom of Figure 2. During the first two periods it was once every 3 h, during the fourth period it was once every 1 h. During the third period the frequency varied and was once every 1 h during 17 April–31 July 2014 and 15 April–9 July 2015, once every 3 h during 1 August–31 October 2014 and once every 12 h from 1 November 2014 to 14 April 2015. The strings installed in 2012, 2013 and 2014 contained up to 128 thermistors covering depth from 0.5 to 12 m with a vertical separation varying from 0.25 to 2 m (Marchenko et al., 2017b; see Figure 2 therein). In 2015 the string contained 31 sensors separated by 0.25–1 m covering a similar depth interval.

### 2.2.2 Data post-processing

Several post-processing routines were applied to the measurements of the subsurface temperature evolution. Firstly, the individual thermistors were calibrated against 0 °C by applying an offset defined as the mode (most frequent value) of the values

measured during July 1–September 1. During this period subsurface temperature is most likely to reach $0\,^{\circ}\mathrm{C}$. The value serves as a natural upper limit for the snow and firn temperature and is well interpreted from plots of measured values against time.

Secondly, spline interpolation was applied to interpolate the data from each thermistor string to a common vertical grid with 0.1 m spacing between neighboring nodes. During April 2012–2015 the subsurface temperature was simultaneously measured by several thermistor strings and thus the interpolated dataset was subsequently averaged horizontally to produce a single series $\tau$ describing subsurface temperature changes in time (superscript $n$) and depth (subscript $i$):

$$\tau_i^n = \frac{1}{q} \sum_{m=1}^{q} \tau_{im}^n, \tag{1}$$

where $\tau_{im}^n$ is the temperature measured by the $m$-th thermistor string and $q$ is the total number of strings. In 2015 only one thermistor string was installed and thus no averaging was applied. To characterize the spread in temperature values measured at the same depth and time but by different instruments the corresponding standard deviations were calculated as

$$(\sigma_{\tau i}^n)^2 = \frac{1}{q-1} \sum_{m=1}^{q} (\tau_{im}^n - \tau_i^n)^2. \tag{2}$$

## 3  Modelling

The evolution of subsurface temperature in the upper 10–15 m of a glacier is mostly controlled by two processes: conductive heat flow and refreezing of liquid water accompanied by release of the latent heat (Cuffey and Paterson, 2010, page 403). Configuring the computational domains to minimize the influence of non-conductive heat fluxes, we compute the effective thermal conductivities for the firn profile at Lomonosovfonna and assess their sensitivity to errors in empirical data used in the simulations.

The computational procedure is as follows. Within the forward model (see Sect. 3.1) the heat equation is approximated numerically and then solved for the temperature with a given conductivity $k$ and density $\rho$. The two parameters are then iteratively adjusted to derive conductivity $k_*$ and density $\rho_*$ minimizing the difference between the simulated and measured subsurface temperature and density (see Sect. 3.2). To quantify the uncertainties associated with $k_*$ we first define the feedback of simulated temperature to change in individual conductivity values. These results are then used to compute the sensitivity of conductivity to errors in the empirically derived:

- temperature values (see Sect. 3.3.1),

- in depths of temperature values (see Sect. 3.3.2) and

- density values (see Sect. 3.3.3).

These routines were coded using MATLAB R-2018a software. All computations on a laptop with four cores Intel(R) i7-4600 CPU 2.1GHz are obtained in less than 10 s.

The time periods are chosen to minimize the influence of water refreezing on the evolution of subsurface temperature. They are referred to as "spring 2012", "spring 2013", "spring 2014", "fall 2014" and "spring 2015" and cover correspondingly 21

April – 19 June 2012, 22 April – 1 June 2013, 18 April – 4 July 2014, 25 September 2014 – 11 April 2015 and 15 April – 9 July
2015. Furthermore, temperature above -2 °C are excluded from the analysis to avoid the influence of latent heat fluxes from
firn volumes with increased water content as the freezing front propagates through them. To minimize the potential influence
of near surface processes (radiation penetration, wind effects) on the results, we disregard the temperature values measured
above the depth of 1 m referenced to the glacier surface at the moment of instrument installation.

### 3.1   Conduction model setup

The model is based on Fourier's law of heat conduction. The temperature of the firn $T$ (°C) at depth $z$ (m) and time $t$ (s) is
governed by the one dimensional equation:

$$\rho(z)\, C\, \frac{\partial T}{\partial t} = \frac{\partial}{\partial z}(k(z)\frac{\partial T}{\partial z}), \tag{3}$$

where $\rho$ is the subsurface density $(\mathrm{kg\, m^{-3}})$, $C = 2027\ \mathrm{J\,(kg\,K)^{-1}}$ is the specific heat capacity calculated using the temperature
dependent function from (Cuffey and Paterson, 2010, page 400) for the temperature of -10°C, and $k$ is the effective thermal
conductivity $(\mathrm{W\,(m\,K)^{-1}})$. Given $k(z)$, $\rho(z)$ along with initial and boundary conditions, Eq. (3) is solved forward in time for
$T(z,\ t)$.

The numerical solution of the Eq. (3) is based on a discretization with the time step $\Delta t$ and space step $\Delta z$. The spatial
and temporal derivatives are approximated using the Crank-Nicolson method (Dahlquist and Björck, 2003) with central finite
differences in space and trapezoidal rule in time. Let $T_i^n$ be the temperature $T(z_i, t^n)$ at $t^n = n\Delta t$, $n = 1,\ldots,N$ and $z_i = i\Delta z$, $i = 1,\ldots,M^n$. The number of time steps is $N$ and the number of nodes in space $M^n$ varies in time. The solution at $z_i$ is
advanced in time from $t^n$ to $t^{n+1}$ as:

$$\rho_i C \frac{T_i^{n+1} - T_i^n}{\Delta t} = \frac{1}{2}\left(\mathbf{D}T_i^{n+1} + \mathbf{D}T_i^n\right),\ n = 1,\ldots,N-1, \tag{4}$$

where $\mathbf{D}T_i^n$ is the spatial temperature derivative at $z_i$ approximated by:

$$\mathbf{D}T_i^n = \frac{(k_{i+1} + k_i)(T_{i+1}^n - T_i^n) - (k_i + k_{i-1})(T_i^n - T_{i-1}^n)}{2\Delta z^2}. \tag{5}$$

Collecting terms in (4) with $T^{n+1}$ on the left hand side and terms with $T^n$ on the right hand side, a tridiagonal system of linear
equations can de derived and solved for $T_i^{n+1}$ by a direct method. The time integration is unconditionally stable and second
order accurate in space and time (Dahlquist and Björck, 2003).

At each time step the computations are performed for $M^n$ nodes from the top at $z_1 = 1$ m to the bottom at $z_{M^n}$ where the
temperature is just below -2°C, which minimizes the influence of the latent heat release at the freezing front. Thus the depth
of the lower boundaries of the simulation domains $z_{M^n}$ increases over time following the downward propagation of the -2°C
isotherm as shown by the white curves in Figure 2. The vertical step $(\Delta z)$ follows the depth interval in the empirical dataset
and is 0.1 m. The chosen time step is 1 h and in case the measurement period is larger than that, linear interpolation is used to
derive the values missing in the upper and lower boundary conditions. The model is initialized in each of the five time periods

using measured temperature values $\tau_i^1$. The upper and lower boundary conditions are of Dirichlet type and are determined by the temperatures measured at $z_1$ and $z_{M^n}$: $T_1^n = \tau_1^n$ and $T_{M^n}^n = \tau_{M^n}^n$.

The density $\rho(z)$ and effective thermal conductivity $k(z)$ at depths $z_i$ are constrained using piecewise linear interpolation based on $J$ nodes vertically spaced by 1 m. Since the forward model is used within an inversion routine (see section 3.2) optimizing the $J$ 1-m spaced $\rho$ and $k$ values, the latter are given by an arbitrary initial guess at the first inversion iteration and later by the results of the previous inversion iteration. For the domains covering spring seasons $J$ equals 8 and is 6 in fall 2014 with the uppermost value corresponding to the upper boundary of the computational domain. The choice of $J$ value is a compromise between coarse vertical resolution of the optimized parameters (low $J$) and insufficient amount of data to constrain a very detailed $k_*$ profile (high $J$). Too large $J$ will result in an oscillatory optimal solution for $k$.

## 3.2 Inverse routine

The effective thermal conductivity $k$ in (5) is unknown. Therefore, the above described forward model is used in an optimization routine to iteratively derive the values of the effective thermal conductivity and density that minimize objective function $F_{\tau,\varrho}(k,\rho)$. Following Smith (2013), the latter is defined by the sums of squared differences between the simulated and measured temperature and density:

$$F_{\tau,\varrho}(k,\rho) = \sum_{n=1}^{N} \sum_{i=1}^{M^n} \frac{1}{(\sigma_{\tau_i^n})^2} \left(T_i^n(k,\rho) - \tau_i^n\right)^2 + \gamma \sum_{i=1}^{M^n} \frac{1}{(\sigma_{\varrho_i})^2} \left(\rho_i - \varrho_i\right)^2, \tag{6}$$

and the optimization routine attempts to minimize $F_{\tau,\varrho}(k,\rho)$ by adjusting the conductivity $k$ and density $\rho$ values used in simulations. In eq. (6) $T_i^n(k,\rho)$ and $\tau_i^n$ are the simulated and measured temperature values at time $t^n$ and depth $z_i$, $\varrho_i$ is the density measured at depth $z_i$. For the spring domains $\varrho$ profiles are taken from the cores drilled within a couple of days from the start of simulation and for the fall 2014 domain we reuse the density data from the core drilled in April 2014. The deviations of the simulated temperature values from empirical data are weighted by the variances $(\sigma_{\tau_i^n})^2$ in temperature values from different thermistor strings but at the same time and depth (see Eq. (2)). This results in lower significance of simulation errors when the measurements are less certain. It is thus assumed that measurement errors are independent and normally distributed with zero mean and variance $(\sigma_{\tau_i^n})^2$. Since only one shallow core was drilled every year, empirical data are not available to quantify the errors in density measurements, the weighting term $\sigma_{\varrho_i}$ was set to $1\,\mathrm{kg\,m^{-3}}$. The value $\gamma = 10$ is chosen to keep the balance between the temperature and density terms in $F$ such that the optimal solutions are smooth and $\rho$ is close to the measurements in $l^2$-norm. The choice of $\gamma$ depends on the size of the data $N$, $M^n$ as well as the magnitudes of $\tau$ and $\varrho$.

Choice of the cost function in (6)) does not assume any correlation between $k$ and $\rho$. This relation is derived later in Section 4.6 based on the optimized $k$ and $\rho$ values. The primary aim here is to derive the optimal $k$ values at $J$ nodes. Optimization of the densities used by the forward model is included in the optimization to allow for flexibility in the parameter $\rho$, since the measurements in single firn cores are uncertain and may be not representative at the scale ca 10 m covered by thermistor measurements. The second term on the right hand side of equation (6) can be interpreted as a Bayesian prior guess (Smith, 2013; Calvetti and Somersalo, 2007) adding extra information about $\rho$ or a regularization of the density according to Tikhonov (Calvetti and Somersalo, 2007).

Introduce the vectors:

$$T = (T_1^1, T_1^2, \ldots, T_1^N, T_2^1, \ldots, T_2^N, T_3^1, \ldots, T_M^N)^T,$$
$$\tau = (\tau_1^1, \tau_1^2, \ldots, \tau_1^N, \tau_2^1, \ldots, \tau_2^N, \tau_3^1, \ldots, \tau_M^N)^T,$$
$$\rho = (\rho_1, \rho_2, \ldots, \rho_M)^T,$$
$$\varrho = (\varrho_1, \varrho_2, \ldots, \varrho_M)^T,$$

where $M = \max_n M^n$. Let the diagonal matrices $W$ have $W_{jj} = 1/(\sigma_{\tau_i^n})^2$, $j = N(i-1) + n$, $n = 1, \ldots, N$, $i = 1, \ldots, M^n$, and $W_{jj} = 0$, $j = N(i-1) + n$, $n = 1, \ldots, N$, $i = M^n + 1, \ldots, M$, on the diagonal. Then $F$ in Eq. (6) can be written as:

$$F_{\tau,\varrho}(k, \rho) = (T - \tau)^T W (T - \tau) + \gamma (\rho - \varrho)^T (\rho - \varrho). \tag{7}$$

The diagonal elements of $W$ vanish when $i > M^n$ since the sums in Eq. (6) are restricted to $i \leq M^n$.

The optimal $k_*$ and $\rho_*$ minimize $F$ in Eq. (6) and (7) and the solution of the nonlinear least squares optimization problem can be written as:

$$(k_*, \rho_*) = \arg\min_{k,\rho} F_{\tau,\varrho}(k, \rho). \tag{8}$$

It is solved by the **lsqnonlin** function in MATLAB.

## 3.3 Uncertainty quantification

Results of the optimization routine largely rely on the empirical data used to guide the routine. The sensitivity of optimized $k_*$ values is explored separately for the errors in measured temperature, depths of empirically derived temperature values (both affecting the vertical temperature gradient) and measured density values by applying uncertainty quantification techniques described in (Smith, 2013). The results of the sensitivity estimates are only valid for relatively small errors on the order of *ca* $5 - 10$ % of the parameter value in question.

### 3.3.1 Temperature

Errors $\delta\tau$ in the measured temperature values $\tau$ in Eq. (6) propagate to the effective thermal conductivities $k_*$ derived from the optimization problem in Eq. (8). Here we first inverse the logic and calculate the feedback of simulated temperature values $T$ to relatively small perturbations in individual $k_*$ values. These results are then used to define the response of optimized thermal conductivities to possible errors in temperature data.

In general, temperature deviations resulting from perturbation $\delta k$ in the effective thermal conductivity vector $k_*$ can for time $t^n$ and depth $z_i$ be described by the equation:

$$T_i^n(k_* + \delta k, \rho_*) - T_i^n(k_*, \rho_*) = \sum_{j=1}^{J} A_{ij}^n \delta k_j, \tag{9}$$

where $A_{ij}^n$ represent the local temperature responses at $t^n$ and $z_i$ to unit perturbations in $k_j, j = 1, \ldots, J$. The full sensitivity matrix $A$ describes the spatio-temporal distribution of the deviations between temperature simulations done using perturbed

and not perturbed effective thermal conductivities. $A$ is defined as:

$$A = \begin{matrix} A_{11}^1 & A_{12}^1 & \cdots & A_{1J}^1 \\ \vdots & & & \vdots \\ A_{11}^N & A_{12}^N & \cdots & A_{1J}^N \\ A_{21}^1 & A_{22}^1 & \cdots & A_{2J}^1 \\ \vdots & & & \vdots \\ A_{M1}^N & A_{M2}^N & \cdots & A_{MJ}^N \end{matrix}. \tag{10}$$

Each row corresponds to specific indices in time, $n = 1, \ldots, N$, and space, $i = 1, \ldots, M$ and columns correspond to different $k$ values $j = 1, \ldots, J$ and in a general case $J$ is not fixed to 6 or 8 and can be equal to $M$. Then the matrix form of Eq. (9) is:

$$T(k_* + \delta k, \rho_*) = T(k_*, \rho_*) + A\delta k. \tag{11}$$

To derive the $J$ columns in the sensitivity matrix $A$ in Eq. (10) the forward model (see Eq. (3)) was run $J$ times consecutively perturbing the individual thermal conductivities $k_{*\,j}, j = 1, 2 \ldots, J$ by the value $\Delta k$. The elements of $A$ are approximated by the finite difference formula:

$$A_{ij}^n = \frac{T_i^n(k_* + \delta k_j, \rho_*) - T_i^n(k_*, \rho_*)}{\Delta k}, \tag{12}$$

where the entries of $\delta k_j$ are zero except for the $j$:th one which is $\Delta k = 10^{-6}$ W $(\mathrm{m\,K})^{-1}$. Note that a perturbation of $k_{*j}$ will change all values of $k(z)$ between $z_{j-1}$ and $z_{j+1}$ due to the linear interpolation for $k$. The full matrix $A$ can be used to quantify deviations in the optimal $k_*$ given an assumption regarding the possible errors in measured temperature values. Since the upper and lower boundary conditions are given by $\tau_1^n$ and $\tau_{M^n}^n$ for every $k$, the elements $A_{1j}^n$ and $A_{M^n j}^n$ for $j = 1, \ldots, J$, $n = 1, \ldots, N$, in $A$ in Eq. (10) are zero. Consequently, we do not assess the effect of errors in the temperature measurements on $k_*$ through the values used to initialize and force the forward model.

If we now assume that $\delta\tau$ are perturbations in the temperature data $\tau$, that will result in deviations $\delta k$ of the optimized thermal conductivity values from the original estimate $k_*$. The optimization problem in Eq. (8) now has the solution:

$$k_* + \delta k = \underset{k}{\arg\min}\, F_{\tau + \delta\tau, \varrho}(k, \rho_*). \tag{13}$$

If $\delta\tau$ in Eq. (13) is small and $\rho_*$ is fixed, then using Eq. (7) and Eq. (11) one can show that $\delta k$ in Eq. (13) satisfies:

$$A^T W A \delta k = A^T W \delta\tau. \tag{14}$$

By linearising the dependence of $T_j(k_* + \delta k, \rho_*) - T(k_*, \rho_*)$ on $\delta k$ in Eq. (11), we arrive at a linear least squares problem to solve for $\delta k$ in Eq. (14):

$$\delta k = (A^T W A)^{-1} A^T W \delta\tau = A_\tau \delta\tau. \tag{15}$$

The weight matrix $W$ in Eq. (15) is used to select only those values of $\delta\tau$ that are inside the computational domain. Columns in the sensitivity matrix $A_\tau$ correspond to the different $k_{*j}$ nodes and individual element express the feedback to be expected from the respective thermal conductivity value to unit perturbations at different depths and moments in time.

30    It is further assumed that errors in temperature measurements $\delta\tau$ are independent in space and constant in time and can be expressed as:

$$\delta\tau = \mathcal{E}\delta\varsigma, \tag{16}$$

where the elements of column vector $\delta\varsigma$ are normally distributed random variables $\delta\varsigma_i \sim \mathcal{N}(0, s_i^2)$, $i = 1, \ldots, M$, with zero mean and standard deviations $s_i$. The elements of the $MN \times M$ matrix $\mathcal{E}$ are zero except for $\mathcal{E}_{ki} = 1$, $k = N(i-1) + n$, $n =$

$1, \ldots, N$, $i = 1, \ldots, M$. Then the error in $\tau_i^n$ is $\delta\tau_i^n = \delta\varsigma_i$. The expected values of $\delta k$ can be evaluated using Eq. (15) and Eq. (16):

$$\mathbb{E}[\delta k] = A_\tau \mathcal{E}\mathbb{E}[\delta\varsigma] = 0, \tag{17}$$

and the variances are found on the diagonal of the covariance matrix

$$
\begin{aligned}
\mathbb{C}\text{ov}[\delta k] \quad &= \mathbb{E}[(A_\tau \delta\tau)(A_\tau \delta\tau)^T] = A_\tau \mathbb{E}[\delta\tau\delta\tau^T]A_\tau^T \\
&= A_\tau \mathcal{E}\mathbb{E}[\delta\varsigma\delta\varsigma^T]\mathcal{E}^T A_\tau^T = A_\tau \mathcal{E}S\mathcal{E}^T A_\tau^T,
\end{aligned}
\tag{18}
$$

where $S$ is a square, diagonal $M \times M$ matrix with $s_i^2$ on the diagonal. In the numerical experiments for the first four time periods when nine thermistor strings were used $s_i$ is taken as the time average of $\sigma_{\tau_i^n}$ in Eq. (2): $s_i^2 = \frac{1}{N}\sum_{n=1}^N \sigma_{\tau_i^n}^2$ and for spring 2015 when only one thermistor string was used as a vertically uniform value of $s_i^2 = 0.02$.

### 3.3.2   Depth of temperature sensors

The process of conductive heat flow is governed by the spatial temperature gradient $\frac{\partial T}{\partial z}$. Uncertainty in the empirical estimates

of the gradient depends not only on the accuracy and precision of the measured temperature values, but also on the possible biases in the estimates of the depths of the sensors. Real positions of the latter may differ from the ones assumed according to the original design due to the curvature of the thermistor cable in the borehole. Here possible biases in $z$ are converted to possible biases in $T$ using the definition of the temperature gradient and the uncertainty in $k_*$ is quantified using $A_\tau$ in Eq. (15).

The error in vertical separation of the temperature values attributed to depths $z_i$ and $z_{i+1}$ is assumed to be constant in time and is denoted by $\delta\epsilon_{i+1}$. Then

$$z_{i+1} = z_i + \Delta z + \delta\epsilon_{i+1} = z_1 + i\Delta z + \sum_{j=1}^{i+1} \delta\epsilon_j = z_1 + i\Delta z + \delta z_{i+1}, \tag{19}$$

where $i = 1, \ldots, M^n - 1$, $\delta z_i$ is the cumulative error in depth. At $z_1$ we have the error $\delta z_1 = \delta\epsilon_1$. If $\delta\epsilon_i$, $i = 1, \ldots, M^n$, are normally distributed random variables independent of each other and with the mean values $\epsilon$ and variances $\sigma_z^2$, $\delta\epsilon_i \sim \mathcal{N}(\epsilon, \sigma_z^2)$,

then $\delta z_i \sim \mathcal{N}(i\epsilon, i\sigma_z^2)$. With the lower triangular $M \times M$ matrix $R$ in

$$
R = \begin{bmatrix} 1 & 0 & \cdots & 0 \\ 1 & 1 & \cdots & 0 \\ \vdots & & & \\ 1 & 1 & \cdots & 1 \end{bmatrix},
$$

the relation between local ($\delta\epsilon$) and cumulative ($\delta z$) depth errors is $\delta z = R\delta\epsilon$.

Temperature perturbation value $\delta\tau_i^n$ can be expressed using the temperature gradient and the position error $\delta z_i$ as

$$
\delta\tau_i^n = \delta z_i \left.\frac{\partial T}{\partial z}\right|_{z=z_i, t=t^n}. \tag{20}
$$

Let the elements of the matrix $\mathcal{T}$ be zero except for $\mathcal{T}_{ki} = \left.\frac{\partial T}{\partial z}\right|_{z=z_i, t=t^n}$, where $k = N(i-1)+n$, $n = 1, \ldots, N$, $i = 1, \ldots M$. Then $\delta\tau = \mathcal{T}\delta z = \mathcal{T}R\delta\epsilon$. Thus, the minimal $\delta k$ in Eq. (13) satisfies Eq. (14) with the solution

$$
\delta k = (A^T W A)^{-1} A^T W \mathcal{T} R \delta\epsilon = A_z \delta\epsilon, \tag{21}
$$

with elements of the matrix $A_z$ expressing the response of the $k_*$ value in question to unit perturbations in depths of empirical temperature values (cf. with $A_\tau$ in Eq. (15)). The expected value of $\delta k$ is

$$
\mathbb{E}[\delta k] = A_z \mathbb{E}[\delta\epsilon] = \epsilon A_z e_M, \tag{22}
$$

where $e_M$ is a vector of length $M$ with elements 1. Since stretching of the cable in a borehole is not likely, both $\delta\epsilon$ and $\mathbb{E}[\delta k]$ values are rather negative than positive.

In line with Eq. (18), the covariance in $\delta k$ is

$$
\mathbb{C}\mathrm{ov}[\delta k] = A_z \mathbb{C}\mathrm{ov}[\delta\epsilon] A_z^T = \sigma_z^2 A_z A_z^T. \tag{23}
$$

The variances of $\delta k$ are found on the diagonal of $\mathbb{C}\mathrm{ov}[\delta k]$

$$
\mathbb{V}\mathrm{ar}[\delta k_j] = \sigma_z^2 \sum_{i=1}^{M^n} A_{zji}^2 \tag{24}
$$

and tell how the variances $\sigma_z^2$ are magnified through $\delta\epsilon$ and to the variances in thermal conductivity estimates.

In the absence of empirical estimates of deviation of the real depths of sensors with respect to the designed ones, the quantification of $\epsilon$ is based on the fact that the vertical deviation of the direction of cable with thermistors is constrained by the
walls of the borehole separated by diameter $d = 0.05$ m. The uncertainty $\epsilon$ can be expressed as:

$$
\epsilon = L - \sqrt{L^2 - d^2}, \tag{25}
$$

where $L$ is the typical separation between two points on the cable where it touches the walls of the borehole. In case $L$ is assumed to be equal to 1 m, $\epsilon$ is 1.3 mm.

### 3.3.3 Density

Both the forward model and the optimization routine rely on the empirical data on the subsurface density. To assess the sensitivity of optimized thermal conductivity values $k_*$ to errors in density data $\varrho$ we first assess the feedback of simulated errors to deviations in $\rho_*$ and then translate these results to perturbations $\delta k$ using the matrix $A_\tau$ (see Eq. (15)).

The sensitivity of the temperature due to perturbations $\delta\rho$ in $\rho$ is computed following the logic of Eq. (9). The sensitivity matrix $B$ is defined as:

$$T_i^n(k_*, \rho_* + \delta\rho) = T_i^n(k_*, \rho_*) + \sum_{j=1}^{J} B_{ij}^n \delta\rho_j, \tag{26}$$

or in matrix form:

$$T(k_*, \rho_* + \delta\rho) = T(k_*, \rho_*) + B\delta\rho, \tag{27}$$

with the elements $B_{ij}^n$ in $B$ calculated and ordered in the same manner as in $A$ in Eq. (10) and (12). A perturbation $\delta\rho$ can be interpreted as a perturbation in the temperature $\delta\tau = -B\delta\rho$ in the first term of (7) and by Eq. (15) it follows that the
perturbation in $k_*$ will be

$$\delta k = -(A^T W A)^{-1} A^T W B \delta\rho = -A_\rho \delta\rho. \tag{28}$$

The second term in Eq. (7) has no influence on the solution here since we minimize over $k$.

### 3.3.4 Hessian of the objective function

The behavior of the objective function $F_{\tau,\varrho}(k, \rho)$ in (6) and (7) close to the optimum $k_*, \rho_*$ is determined by the Hessian matrix
$H$:

$$H = \begin{pmatrix} A^T W A & A^T W B \\ B^T W A & B^T W B + \gamma I \end{pmatrix}, \tag{29}$$

where $A$, $B$, $W$ and $\gamma$ are defined in (11), (27), (7) and (6) and $I$ is the identity matrix. A small perturbation $\delta\chi = (\delta k, \delta\rho)$ of $k_*, \rho_*$ will change $F_{\tau,\varrho}$ by $\delta\chi^T H \delta\chi$. The Hessian $H$ is positive definite with positive eigenvalues and orthogonal eigenvectors when $\gamma > 0$. Thus, $k_*, \rho_*$ is a local minimum of $F_{\tau,\varrho}$. In the direction $\delta\chi$ of an eigenvector with a large eigenvalue, the optimum
is well defined but in a direction along an eigenvector corresponding to a small eigenvalue the uncertainty in the optimum is larger.

## 4 Results and discussion

### 4.1 Measured subsurface density and temperature

The subsurface glacier profile observed in the cores consists of snow and firn with multiple ice lenses (Figure 1). The measured
snow and firn density ($\varrho$) varies from 350 to 900 $\mathrm{kg\,m^{-3}}$ with a gradual increase downward and occasional spikes corresponding

to the ice layers apparent from the stratigraphical record. Compared with the measured values, the optimized densities ($\rho_*$) show similar ranges and the overall pattern of an increase with depth (Figure 1).

The evolution of subsurface temperature $\tau_i^n$ measured during five periods used for the simulations is shown in Figure 2. The position of the upper boundary shifts upward following the snow accumulation at the surface. The evolution of deep temperature outside of the melt season generally follows the surface temperature with significant dampening with depth and time delay of the amplitude. During fall 2014 temperature continuously decreases at all depths. During the four spring seasons the same pattern is observed only below $ca$ 3-4 m, while the upper part of the profile experienced warming. The measured temperature generally increases with depth, however, the opposite tendency is observed for the upper $ca$ 1-2 m of the profile in the spring seasons, particularly towards the end of the simulation periods. The simulation domains where Eq. (3) is solved and $k$ and $\rho$ are optimized in Eq. (8) are bound by the white curves limiting the area with temperature values colder than -2°C and depths at the time of instrument installation larger than 1 m. The snow and firn temperature measured in April–May 2014 is significantly lower than the values registered at similar depths below the surface in April–May of the following year (Figure 2). Based on this finding we suggest that the late part of the winter season in 2014 (February–April) was colder than in 2015. This finding is supported by the data from an AWS at Nordenskiöldbreen (600 m asl), according to which in 2015 March and April were warmer than in 2014 by 2.5 and 6°C correspondingly (description of AWS and data can be found at: http://www.projects.science.uu.nl/iceclimate/aws/). The vertical temperature gradients measured in 2015 on April 11 and 15 by the thermistor strings installed a year earlier and by the new installation are in good correspondence (Figure 2). This suggests that during one year (April 2014–April 2015) gravitational densification of snow and firn did not result in significant change of the separation between sensors and justifies the usage of time-constant density profiles in our simulations.

## 4.2 Optimized thermal conductivity values $k_*$

The effective thermal conductivities $k_*$ optimized according to the Eq. (8) range from 0.2 to 1.6 W $(\mathrm{m\,K})^{-1}$ and are presented in Figure 3B. The values consistently increase with depth at the rate of $ca$ 0.11 W $(\mathrm{m^2\,K})^{-1}$ on average and somewhat slower in the fall of 2014. The temporal change in $k_*$ values can also be assessed with reference to Figure 3B since the profiles for different domains are plotted with a vertical offset accounting for the surface accumulation. The overall tendency is increase in $k_*$ over time with an average rate of 0.09 W $(\mathrm{m\,K\,year})^{-1}$. Provided that the surface accumulation rates at the field site are slightly above 1 m, this is less than the expected value from the vertical gradient in $k_*$. In the absence of a physical process that could result in a decrease of the firn effective thermal conductivity $k_*$ over time, the apparent lowering of optimized $k_*$ values, such as seen between 8 and 10 m from spring 2012 to spring 2013, is attributed to the uncertainties in empirical data such as temperature readings, depths of individual sensors, density measurements and estimates of the surface accumulation rate.

The uncertainty bar for each $k_*$ value is calculated following Eq. (18) and denotes the intervals of one standard deviation from the mean value. The bars take into account only the possible errors in subsurface temperature measurements quantified through the time-averaged standard deviations in values from different thermistor strings (Figure 3A). Compared with the $k_*$, the uncertainty values are generally small and are between $ca$ 0.03 and 0.25 W $(\mathrm{m\,K})^{-1}$. The overall increase in uncertainty of $k_*$ values with depth is due to the downward decrease in temporal and spatial temperature gradients and also in the number

of measurements with temperature below -2 $^{\circ}$ C. Altogether this results in increase of chances for the optimization routine to converge to a "wrong" $k_*$ value based on biased temperature simulations resulting from it, as the biases are smaller and lesser in number than in the upper part of the profile. The deepest $k_*$ values in spring 2012 and 2013 exhibit outstandingly high uncertainty values, which is explained by the fact that the values are found at depths where temperature never reaches below -2 $^{\circ}$C and are thus constrained only through the part of the linear fit to the $k_*$ value above that lies within the simulation domain. Due to the lack of data and large variances of the temperature measurements deep down, the deepest $k_*$ values in each of the two domains in 2012 and 2013 are likely not reliable.

In Figure 4, the optimal $k_*$ (**A**) and $\rho_*$ (**B**) values are computed for the spring 2012 domain for different $\gamma$. The result varies for the extreme values of $\gamma$ but is consistent for intermediate $\gamma$. The $k_*$ and $\rho_*$ values are indistinguishable for $\gamma = 10, 10^3$ and $\gamma = 10, 10^3, 10^7$, respectively. The eigenvector for the smallest eigenvalue (Figure 4**C**) shows that the uncertainty is largest for the deepest $k_*$ values when $\gamma \geq 10$ in agreement with Figure 3**B**. When $\gamma = 10^{-5}$ then the regularization of $\rho$ is insufficient with oscillations in the solution and an eigenvector with large entries and uncertainty for all $\rho$ values. The results for other seasons are similar.

## 4.3 Sensitivity of $k_*$ to errors in temperature measurements

The results of sensitivity experiments exploring the feedback ($A_\tau$ in Eq. (15)) of $k_*$ values to errors in subsurface temperature measurements $\tau_i^n$ are presented in Figure 5. The black markers close to the right border of each dataset show the positions of $k_*$ points. The color at any specific point in depth $z_i$ and time $t^n$ corresponds to the response of the $k_{*\,j}$ value at the depth indicated by the black circle marker to a unit change in temperature (1 $^{\circ}$C) at depth $z_i$ and time $t^n$. The sensitivity is set to zero outside of the computational domain, the lower boundary of which is shown by the black curves.

It can also be noted that here we analyse the $k_*$ feedback to errors in individual temperature values within the dataset used to constrain the optimization routine. For the first four calculation domains (spring 2012-2014 and fall 2014) this data are the result of lateral averaging of data from nine ($q = 9$ in Eq. (1)) thermistor strings. Although the strings are coupled to the same data logger, the errors in temperature measurements can be assumed to be at least partly independent. Thus sensitivities $A_\tau$ to temperature errors coming from single thermistor strings and not laterally averaged datasets can be expected to be lower than indicated by color coding in Figure 5 by a factor of $\sqrt{q} = 3$.

Optimized $k_*$ values are not very sensitive to single errors in subsurface temperature data: for $j = 1 \ldots 7$ (panels **A** - **G** in Figure 5) the expected response of the thermal conductivity values $k_{*j}$ to a 1 $^{\circ}$C error varies between $-1.1 \cdot 10^{-3}$ and $1.1 \cdot 10^{-3}$ W $(\mathrm{m\,K})^{-1}$, corresponding to the $\bar{A}_{\tau j} - 3\sigma_{A_{\tau j}}$ and $\bar{A}_{\tau j} + 3\sigma_{A_{\tau j}}$, where $\bar{A}_{\tau j}$ and $\sigma_{A_{\tau j}}$ are the mean and standard deviation of the $A_{\tau j}$ values. However, as it was demonstrated earlier (see the error bars in Figure 3B), a systematic time-independent bias in temperature data can result in notable deviations of the $k_*$ estimates.

Sensitivities $A_\tau$ for the fall 2014 domain exhibit a distinctively different range and pattern of spatiotemporal change of values with respect to the spring seasons. The reasons for that are not completely understood and elucidation may require additional empirical data from other fall seasons. It can be noted that during the period from September 2014 to April 2015

the dominant tendency in the change of surface temperature was decrease, which induced continuous cooling of the subsurface profile.

Several patterns in the spatiotemporal distribution of the sensitivities $A_\tau$ can be noted. In most cases the sign of the sensitivity is positive between the depths of $k_{*\ j-1}$ and $k_{*\ j}$ and negative between the depths of $k_{*\ j}$ and $k_{*\ j+1}$ if $k_{*\ j}$ is the $k$ value being tested for sensitivity (depth is marked by the black circle in Figure 5). This pattern is reversed when considering depth levels further away from the circles (between $k_{*\ j-2}$ and $k_{*\ j-1}$ and between $k_{*\ j+1}$ and $k_{*\ j+2}$) and the period of alternation roughly corresponds to the spacing between $k_*$ values which is 1 m. This result is explained through changes in the vertical temperature gradient induced by the temperature perturbations. The overall pattern in the vertical change of temperature is to increase downwards, thus a temperature increase at a certain depth results in increase of the temperature gradient just above and decrease just below that depth. These changes in temperature gradient are respectively compensated by negative and positive deviations in $k$ values. Due to the piecewise-linear interpolation of $k$ profile based on $J$ $k_*$ values, a perturbation in $k_{*j}$ also affects the thermal conductivities below and above it. Therefore $k_{*\ j-1}$ and $k_{*\ j+1}$ adjacent to $k_*$ will display the tendency opposite to the one demonstrated by $k_{*\ j}$ to compensate for the associated changes in the heat flux. This pattern is also apparent when comparing different panels in Figure 5.

In most cases $k_*$ is most sensitive to temperature errors *ca* $0.5 - 1$ m above and below its evaluation level, which is evidenced by the more intense colors in the sensitivity fields found in vicinity of the black circle markers. The amplitude of cycles with alternating sign demonstrated by the sensitivity $A_\tau$ fades away with distance from the perturbed $k_*$. The $k_*$ values found deeper down in the vertical profile appear to be more sensitive to errors in temperature, as is seen in more intensive colors around the circular markers in panels **E — H** of Figure 5 compared to panels **A — D** in the same figure.

### 4.4   Sensitivity of $k_*$ to errors in depths of temperature measurements

The feedback of optimized thermal conductivities to errors in depths of temperature values used to constrain the optimization routine ($A_z$ in Eq. (21)) is presented in Figure 6, where different panels show results for the five simulation domains. The color of a point corresponding to index $j$ and depth $z$ indicates the expected bias in $k_{*\ j}$ found at depth highlighted by the black circle in column $j$ given a negative bias of -1 cm in depth of temperature values at depth $z$. The sensitivities $A_z$ for the lowermost $k_{*j}$ values ($j = 8$ for the spring domains and $j = 6$ for the fall 2014) are significantly larger than for other $k_{*j}$ nodes due to propagation of the corresponding high $A_\tau$ values (Figure 5 G, H) to the $A_z$ matrices as is shown in Eq. (15) and (21). These results are not shown.

In Figure 6 blue anomalies in vicinity of the markers indicate that the largest response of $k_{*j}$ can be expected to depth errors just below and just above the depth $z_j$. This pattern is expected since the assumed depth perturbations are negative, which increases the vertical temperature gradient and is compensated by lower $k_*$ values to keep the same heat flux. Secondly, the alternating pattern in the sensitivity of $k_*$, noted in the previous section, can also be seen in the $A_z$ matrices, particularly for the deeper thermal conductivities with larger indices $j$: farther away in the vertical direction from the circular markers the negative anomalies in $A_z$ are replaced by less negative and even positive values and then switch back to significantly negative range. This behaviour of the sensitivity is caused by the piecewise linear interpolation applied to derive the 0.1 m-spaced $k$

profile used in the forward model from 1 m-spaced optimized $k_*$ values. In an attempt to preserve the heat flux and minimize the misfit between measured and simulated measurements the optimization routine will tend to overestimate the $k_{*\ j+1}$ and $k_{*\ j-1}$ values in case $k_{*\ j}$ is forced to be too low due to the perturbation in depths of temperature values. The uncertainties $A_z$ in Figure 6 are notably larger above the circular markers The thermal conductivities are much more sensitive to errors in depths of temperature values occurring above the depth level than below because of the accumulation of the position errors from upper levels to the bottom. Another reason is probably that the vertical temperature gradients are larger in the upper part of the profile. At the same time, due to the influence of the cable weight, thermistor strings can be expected to experience less coiling in the shallow part of a borehole, possibly compensating the larger $A_z$ values there.

Assuming that thermistor strings are in contact with the borehole walls every $L = 1$ m (see Eq. (25)), the mean values of $\delta k$ in Eq. (22) are on the order of -0.02—0.07 W $(\text{m K})^{-1}$ for $k_{*1} - k_{*6}$ and significantly larger lower down in the profile, where temperature gradients are lower and the amount of available empirical data is less (Table 1). The expected uncertainty is 3–8 times larger for the domain covering fall 2014, which is most likely caused by the larger temperature gradients during the early period of subsurface cooling. It can be noted that the uncertainty $\delta k$ in Eq. (22) is scaled by $\epsilon$ in Eq. (25), suggesting that if $L = 0.5$ m, the values in Table 1 will increase almost by a factor of two. The results for the variance in Table 1 using Eq. (24) show how the variance $\sigma_z^2$ in the positional error is magnified as a variance of the error in $k$. If $\sigma_z \approx \epsilon$ then $\sqrt{\mathbb{V}\text{ar}[\delta k]} \approx 0.01$ for most $k$-values, a rather small standard deviation.

## 4.5 Sensitivity of $k_*$ to errors in density

The results of experiments exploring the sensitivity of optimized thermal conductivities to possible errors in density are presented in Figure 7. All panels in the figure are dominated by colors corresponding to the positive sensitivity values, suggesting that the general tendency is an increase in $k_*$ in response to an overestimation of density in the empirical dataset. Negative values correspond to $k_{*j}$ values found in the lower part of the profile ($j = 6\ldots8$), which are generally less reliable.

Thermal conductivity appears to be more sensitive to density errors occurring above the depth level in question. This particularly applies to the biases in $\rho_2$. It is still in the upper part of the subsurface profile were density is relatively low and the relative importance of 1 $\text{kg m}^{-3}$ mistake in density is large. These values are not exceeded by the $A_{\rho 1}$ found above, most probably, because of the constraints imposed by the upper boundary condition on temperature errors: $A_\tau$ is always zero at the upper boundary of the calculation domain.

In the absence of empirical data to quantify possible errors in density, it can be noted that $A_\rho$ values presented in Figure 7 can be scaled by any assumed value $\mathbb{E}[\delta\rho]$ to derive errors $\mathbb{E}[\delta k]$ to be expected from $k_*$ values. It follows that a bias of 50 $\text{kg m}^{-3}$ will result in $k_*$ deviation of up to 0.1 W $(\text{m K})^{-1}$.

## 4.6 Comparison of $k_*$ with earlier published results

The relation between optimized effective thermal conductivities and densities is shown in Figure 8 using markers and the associated linear fit:

$$k = 0.301 \cdot 10^{-2} \rho - 0.724 \tag{30}$$

is illustrated by the thick black line. Also shown are the predictions from published similar fits. The Sturm et al. (1997) approximation relies on multiple needle probe measurements. The Calonne et al. (2011) and Riche and Schneebeli (2013) parameterizations are constrained by $k$ values quantified using numerical modelling of the effective thermal conductivity tensor based on detailed three-dimensional micro tomographic models of snow samples. It can also be noted that all three parameterizations are based on $k$ measured in seasonal snow and the datasets included only a few samples with density above $500\,\mathrm{kg\,m^{-3}}$, while our results are based on measurements in firn with generally higher density.

Almost all $k_*$ values are larger than the effective thermal conductivities predicted by the first two parameterizations and the difference increases for larger densities. At the same time, the linear fit to $k_*$ and $\rho_*$ pairs appears to be closest to the Riche and Schneebeli (2013) parameterization. The latter is based on measurements done on faceted and depth hoar samples developing under strong temperature gradient and resulting in anisotropy of the bulk thermal conductivity with larger $k$ in the vertical direction. At Lomonosovfonna, the faceted crystals developing close to the surface are likely affected by the temperate conditions during the melt season. Below the depth of *ca* 1—2 m the vertical temperature gradients are not as high and the mobility of water vapour in pores is reduced due to higher density, which altogether should limit the development of microstructural anisotropy. At the same time, it can be hypothesized that the preferential water flow in snow/firn, reported from the site (Marchenko et al., 2017b), can result in prominent vertically elongated structures below the surface of Lomonosovfonna, favouring anisotropy at a larger spatial scale and faster heat transfer in the vertical direction. The obvious variability in dependence between $k_*$ and $\rho_*$ values across different domains is a further indication of the fact that the vertical dynamics in $k_*$ are caused not only by the changes in density and proxies describing the snow/firn structure at the scale of processes active in its metamorphism are to be included in the $k = f(\rho)$ functions along with the density (Löwe et al., 2013). At the scale of grains such data can be derived using X-ray tomography (e.g. Kaempfer et al., 2005) and at the scale of $0.1 - 1$ m detailed radar surveys can be used (e.g. Dunse et al., 2008; Marchenko et al., 2017a).

The findings regarding effective thermal conductivity values presented above can also be compared with the results of Giese and Hawley (2015), who applied Fourier analysis to continuous temperature measurements and derived the thermal diffusivity value ($\kappa$) of $25 \pm 3$ m$^2$ year$^{-1}$ in the top 30 m of firn pack Greenland Summit. Based on our optimized effective thermal conductivity ($k_*$) and density ($\rho_*$) values and the specific heat capacity of ice ($C$) at -10 °C (Cuffey and Paterson, 2010, page 400), the thermal diffusivity ($\kappa_*$) is calculated as $\kappa_* = k_* \, (\rho_* \, C)^{-1}$. The resulting $\kappa_*$ values lie in the range from 7.55 to 48.77 m$^2$year$^{-1}$ with the mean value and standard deviations of 25.48 and 7.52 m$^2$year$^{-1}$ respectively, providing a prominent match with the results from Giese and Hawley (2015). It can be noted that quantification of $k_*$ using optimization technique requires a much less extensive dataset in terms of time and depth coverage. Furthermore, since infiltration of melt water in the

summer interrupts the conductive heat exchange in firn at Lomonosovfonna, it is not possible to apply Fourier analysis used by Giese and Hawley (2015) on our data.

## 5 Conclusions

The evolution of subsurface temperature was measured in firn at Lomonosovfonna, Svalbard, using several thermistor strings during April 2012 and July 2015. The data cover five periods when the subsurface profile is at least partly at subfreezing conditions. Combined with the density measurements from four cores it was used to reconstruct the effective thermal conductivity and the density of the firn layers. For that we applied an optimization routine minimizing the mean squared difference between the measured and simulated temperature evolutions and the measured and computed density.

The optimized effective thermal conductivity $k_*$ of the firn pack at Lomonosovfonna varies from 0.4 to 1.05 $\mathrm{W\,(m\,K)}^{-1}$ increasing downwards in a maximal likelihood approach for all the time periods. According to the results of sensitivity analysis, $k_*$ is not very sensitive to systematic temperature offsets. Overestimation of the separation between sensors resulting from possible tortuosity of the cable in the borehole leads to overestimation of the $k_*$ values. Positive deviation in density estimates also result in overestimation of the $k_*$ values, while negative density biases lead to an underestimation of effective thermal conductivity.

The $k_*$ results are notably higher than the $k$ values predicted by widely used empirical parameterizations based on the firn density measurements (Sturm et al., 1997; Calonne et al., 2011) and originally constrained by measurements of $\rho$ and $k$ in seasonal snow. This suggests a possible underestimation of the subsurface heat fluxes by firn models relying on the equations. In regions with climate similar to the one observed in Svalbard this is of particular importance for the cold season as the period of conductive cooling is significantly longer than conductive warming occurring in spring before the onset of melt. Thus the refreezing capacity of the firn pack at Lomonosovfonna by the onset of melt is likely to be be underestimated when simulated using the parameterizations by Sturm et al. (1997) and Calonne et al. (2011).

## 6 Data and code availability

The subsurface density and temperature data is available for download from Marchenko et al. (2019b, a). The computer code used for calculations is available from https://github.com/enigne/HeatConductivity.

## 7 Author contribution

SM - conceptualization of the optimization experiments, collection of empirical data, pilot calculations, graphics. GC - conceptualization of the forward model and optimization routines, software development. PL - conceptualization of the uncertainty experiments and supervision of the software development. VP, RP, WvP, CR - funding acquisition, field operations. All authors participated in the writing and editing of the original draft

## 8 Acknowledgements

This publication is a contribution 92 of the Nordic Centre of Excellence SVALI funded by the Nordic Top-level Research Initiative. Authors appreciate the constructive feedback provided by the Editor and two Reviewers (Henning Löwe and Edwin Waddington), their efforts helped to significantly improve the manuscript. Funding was provided by the Vetenskapsrådet grant 621-2014-3735 (VP), Formas grant 214-2013-1600 (Nina Kirchner of the Stockholm University). Authors also acknowledge the additional funding from the Ymer-80 foundation, Swedish Polar Research Secretariat, Polar Program of the Netherlands Organization for Scientific Research (NWO), Arctic Field Grant of the Research Council of Norway and the Margit Althins stipend of the Royal Swedish Academy of Sciences. Logistical support during field campaigns was provided by the Norwegian Polar Institute and the University Centre in Svalbard (UNIS).

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

**Table 1.** Expected deviations of thermal conductivities ($\mathbb{E}[\delta k]$ in Eq. (22)) and the corresponding standard deviations ($\sqrt{\mathbb{V}\mathrm{ar}[\delta k]}$, see Eq. (23), (24)) given the magnitude of thermistor string coiling in the boreholes is expressed by $L = 1$ m (see Eq. (25)) and $\sigma_z = 1$ cm.

| $\mathbb{E}[\delta k]$ | $k_{*1}$ | $k_{*2}$ | $k_{*3}$ | $k_{*4}$ | $k_{*5}$ | $k_{*6}$ | $k_{*7}$ | $k_{*8}$ |
|---|---|---|---|---|---|---|---|---|
| Spring 2012 | -0.046 | -0.040 | -0.051 | -0.050 | -0.045 | -0.002 | 0.054 | 1.575 |
| Spring 2013 | -0.051 | -0.044 | -0.024 | -0.018 | -0.008 | 0.037 | 0.217 | 0.926 |
| Spring 2014 | -0.021 | -0.042 | -0.028 | -0.045 | -0.028 | 0.010 | 0.111 | -0.103 |
| Fall 2014 | -0.118 | -0.132 | -0.187 | -0.230 | -0.244 | -0.131 | | |
| Spring 2015 | -0.029 | -0.069 | -0.020 | -0.039 | -0.032 | -0.016 | 0.069 | 0.138 |
| $\sqrt{\mathbb{V}\mathrm{ar}[\delta k]}$ | $k_{*1}$ | $k_{*2}$ | $k_{*3}$ | $k_{*4}$ | $k_{*5}$ | $k_{*6}$ | $k_{*7}$ | $k_{*8}$ |
| Spring 2012 | 0.073 | 0.063 | 0.077 | 0.069 | 0.068 | 0.074 | 0.123 | 1.779 |
| Spring 2013 | 0.085 | 0.076 | 0.041 | 0.035 | 0.036 | 0.07 | 0.237 | 1.063 |
| Spring 2014 | 0.042 | 0.063 | 0.051 | 0.065 | 0.047 | 0.061 | 0.146 | 0.196 |
| Fall 2014 | 0.175 | 0.189 | 0.258 | 0.316 | 0.331 | 0.344 | | |
| Spring 2015 | 0.051 | 0.1 | 0.035 | 0.06 | 0.048 | 0.042 | 0.087 | 0.153 |

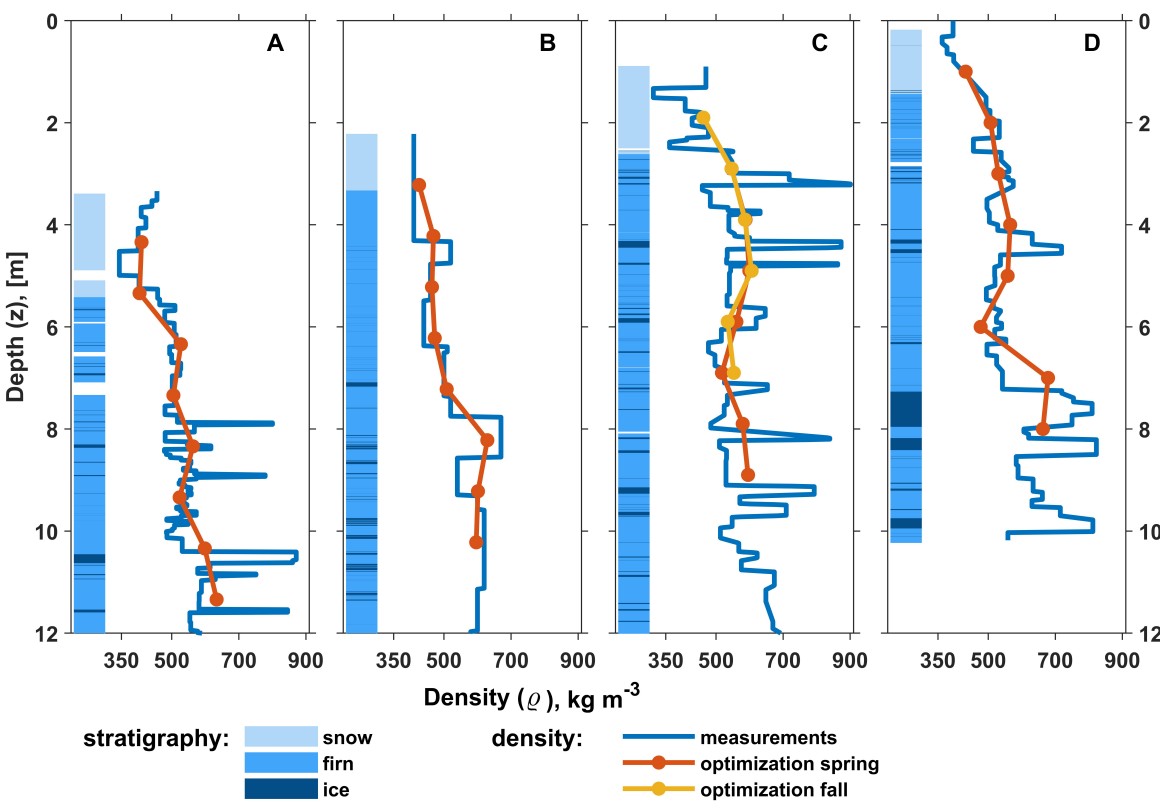

**Figure 1.** Subsurface stratigraphy and density in four cores recovered in April 2012 (A), 2013 (B), 2014 (C) and 2015 (D). Also shown are the density profiles simulated using maximum likelihood approach in Eq. (8).

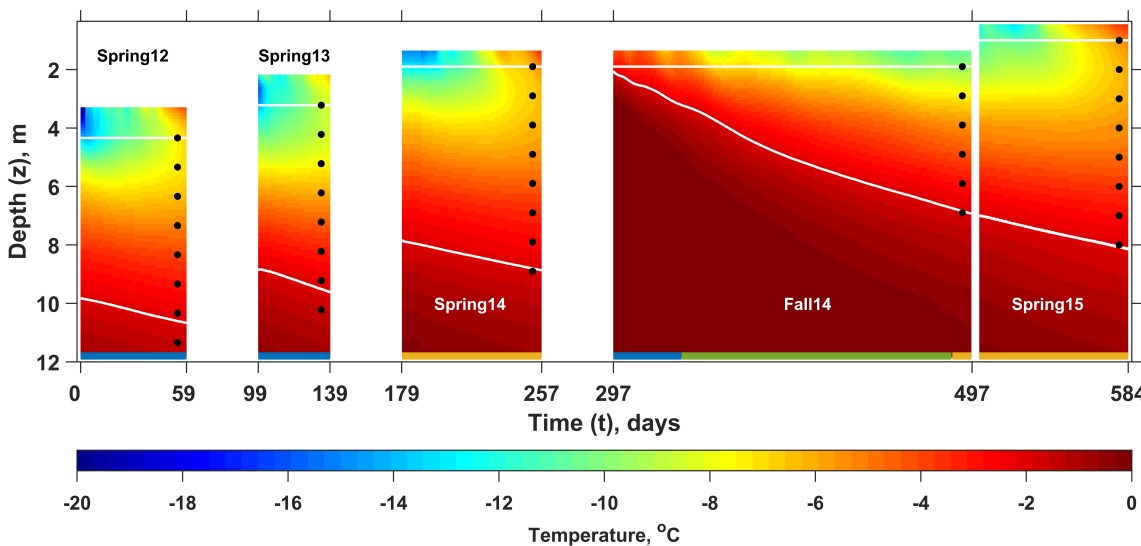

**Figure 2.** Measured evolution of subsurface temperature. The vertical axis is referenced to the glacier surface in April 2015. The horizontal axis is referenced to the $21^{th}$ April 2012 and is linear for the periods covered by measurements: 21 April – 19 June 2012 ("spring 2012"), 22 April – 1 June 2013 ("spring 2013"), 18 April – 4 July 2014 ("spring 2014"), 25 September 2014 – 11 April 2015 ("fall 2014") and 15 April – 9 July 2015 ("spring 2015") and arbitrary between the domains. White curves indicate the upper and lower boundaries of the simulation domains. The color bars along the horizontal axis show the intervals between measurements: yellow - 1 h, blue - 3 h and green - 12 h.

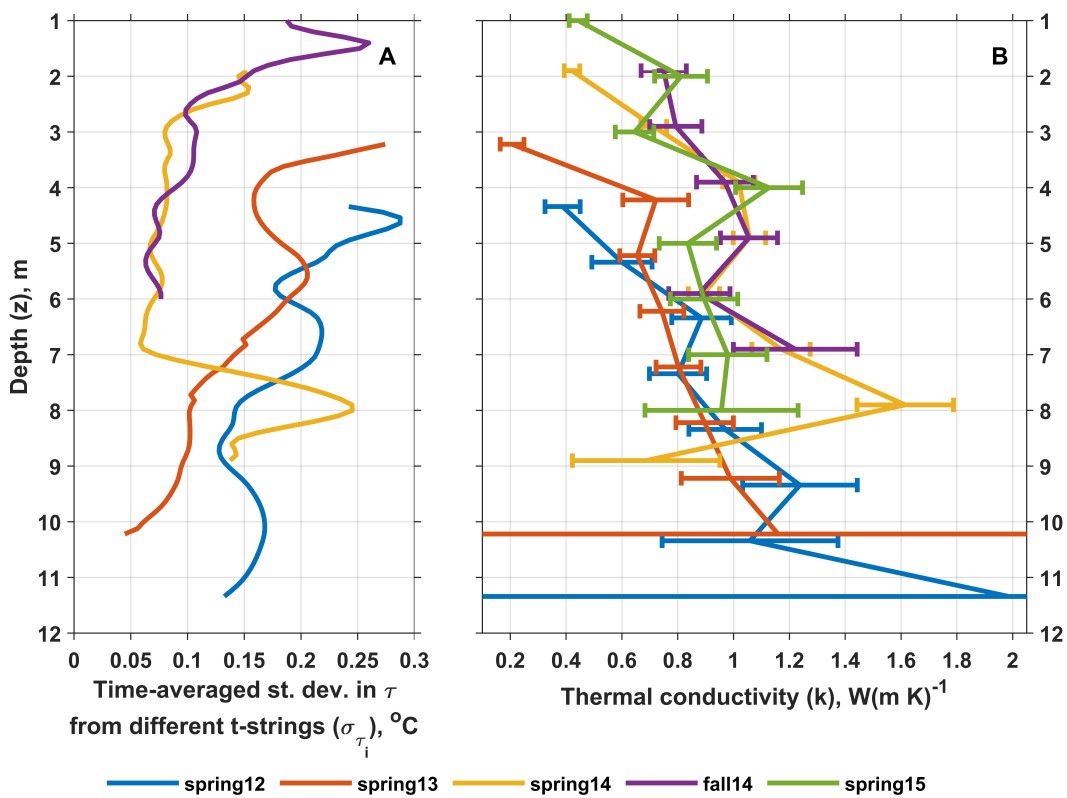

**Figure 3. A**: Time-averaged standard deviations in subsurface temperature measured by different thermistor strings ($s_i$). **B**: Optimized effective thermal conductivity values ($k_*$) calculated following Eq. (8) with error bars according to $\mathbb{C}\text{ov}[\delta k]$ in Eq. (18) assuming temperature errors shown in panel **A**.

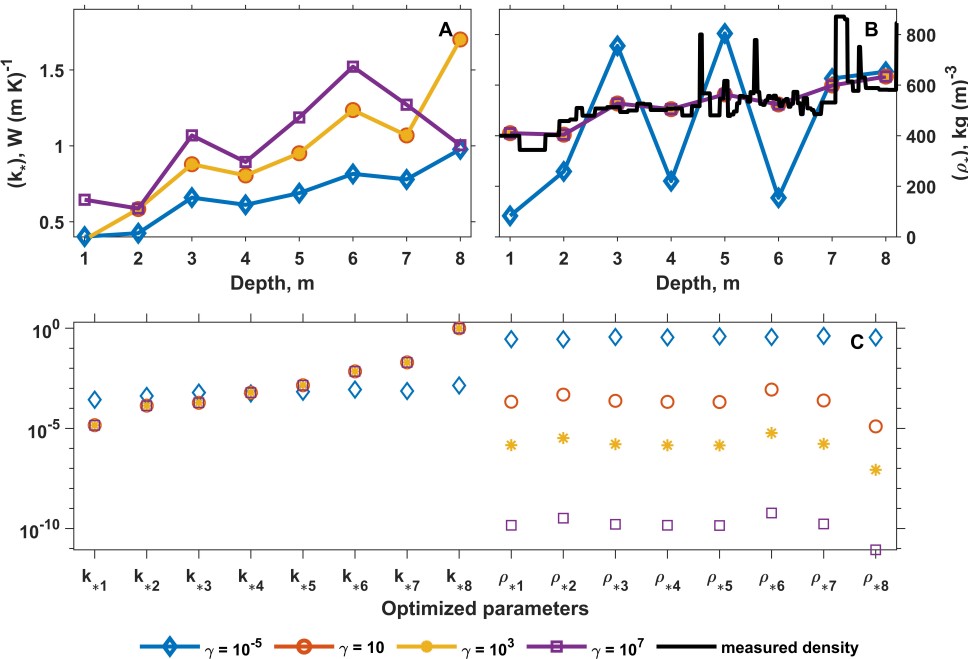

**Figure 4.** The optimal solutions $k_*$ (**A**) and $\rho_*$ (**B**) in the spring 2012 domain for different values of $\gamma$ in equation (7). The black curve in **B** shows measured density values. **C:** absolute values of the eigenvector with the smallest eigenvalue of the Hessian $H$ in equation (29).

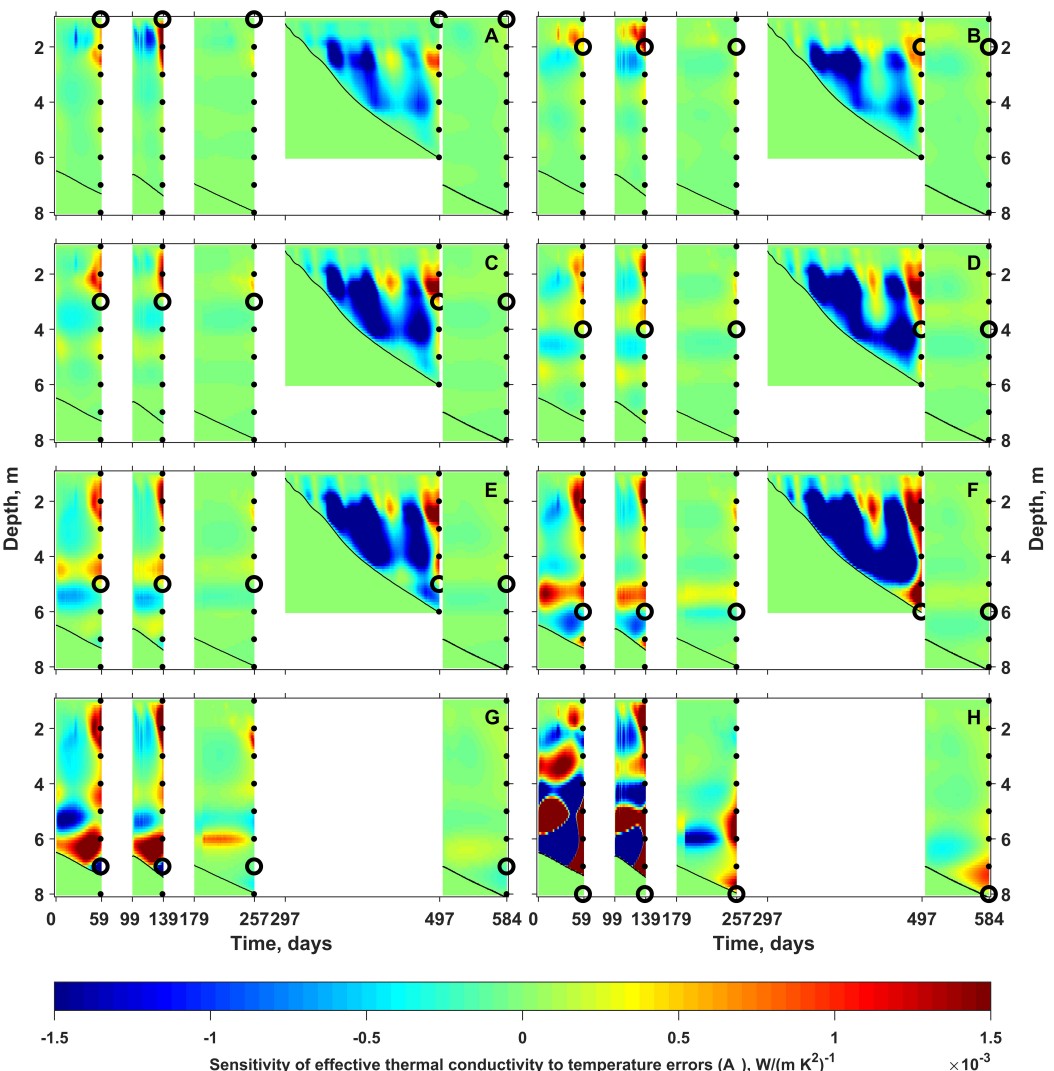

**Figure 5.** Sensitivity ($A_\tau$ in Eq. (15)) of effective thermal conductivities ($k_{*j}$) to errors in the temperature measurements. Panels **A, B - H** correspond to $j = 1, 2, \ldots, 8$, in each case the depth of $k_{*\,j}$ is highlighted by black circle, depths of other $k_*$ values are shown by black dots. The color at a certain point in depth and time corresponds to the feedback of the $k_{*\,j}$ value found at the depth shown by the black circle to a unit error in temperature (+ 1 °C) at that depth and time. Black curves indicate the lower boundaries of the computational domains, below them the sensitivity is set to 0. The horizontal axes are the same as in Figure 2.

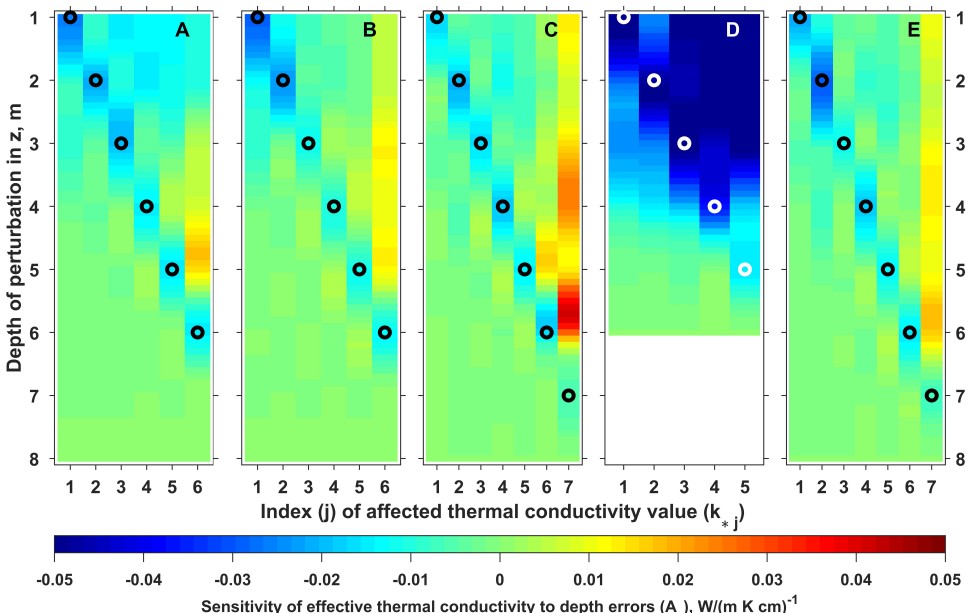

**Figure 6.** Sensitivity ($A_z$ in Eq. (21)) of effective thermal conductivities $k_{*j}$ to errors in depths of temperature values. Panels correspond to different calculation domains: spring 2012 (**A**), spring 2013 (**B**), spring 2014 (**C**), fall 2014 (**D**) and spring 2015 (**E**). In all panels the color of a point in column $j$ and depth $z$ shows the feedback on effective thermal conductivity value $k_{*j}$ found at the depth highlighted by the black (white in **D**) circle to an error in depth of temperature value (1 cm) made at depth $z$.

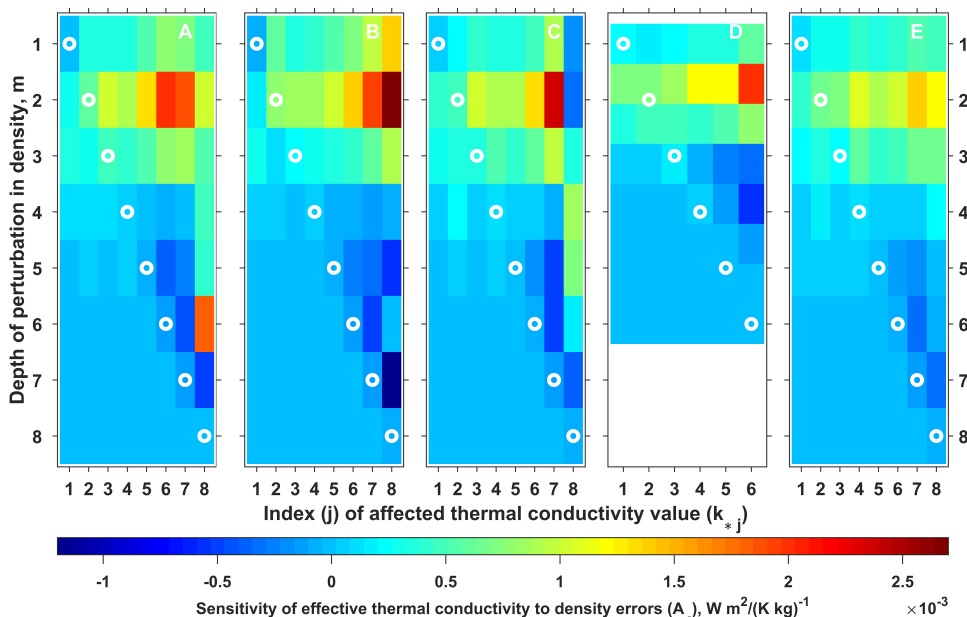

**Figure 7.** Sensitivity ($A_\rho$ in Eq. (28)) of effective thermal conductivities $k_{*j}$ to errors in density. Panels correspond to different calculation domains: spring 2012 (**A**), spring 2013 (**B**), spring 2014 (**C**), fall 2014 (**D**) and spring 2015 (**E**). In all panels the color of a point in column $j$ and depth $z$ shows the feedback of effective thermal conductivity value $k_{*j}$ found at the depth highlighted by the white circle to a unit error in density (1 kg m$^{-3}$) made at depth $z$.

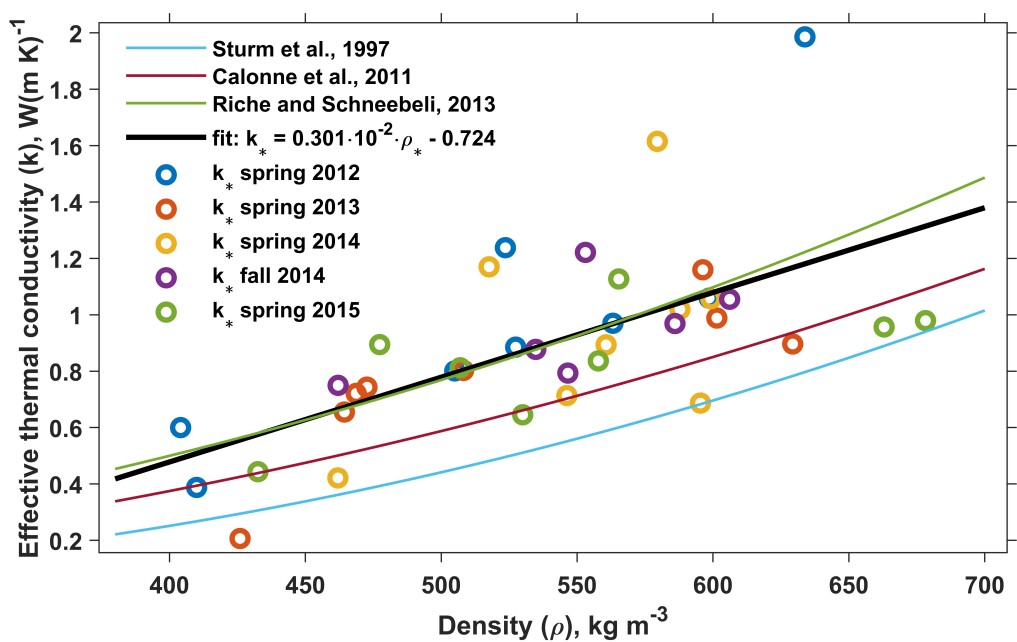

**Figure 8.** Relation between effective thermal conductivities $k$ and densities $\rho$. Markers illustrate the results of optimization routine, the black line shows the linear fit based on all $k_*$-$\rho_*$ pairs. The light blue, ruby and green curves show quadratic $k = f(\rho)$ functions according to (Sturm et al., 1997), (Calonne et al., 2011) and (Riche and Schneebeli, 2013).