# Peer review of "Thermal conductivity of firn at Lomonosovfonna, Svalbard, derived from subsurface temperature measurements"

_The Cryosphere, 2018_

## Referee Comment (RC1) · Henning Loewe (Referee) · 24 Feb 2019

**Review: "Thermal conductivity of firn at Lomonosovfonna, Svalbard, derived from subsurface temperature measurements" by Marchenko et al.**

**General comments**

In the paper the authors retrive profiles of thermal conductivity of near-surface snow/firn in Svalbard as optimization parameters from a comparison of the numerical solution of the heat equation with thermistor (chain) measurements buried over several years. The optimization is restricted to the dry zone.

In my opinion, using intermediate complexity (few parameter) models to constrain physical properties from measurements via optimization is often more conclusive over a mere comparison with more complex (many parameter) models. Even though this method is not new, the work is apparently sound and the authors come to a clear and important conclusion, namely that common density based parametrizations are insufficient for the thermal conductivity in near-surface firn. While I would immediately agree on this finding on physical grounds (see comments), I don't fully understand how the assumption that $\rho$ and $k$ are used as *independent* optimization parameters (which is certainly not true) may affect this finding. A little more discussion seems required here. This and other things are listed in the comments below.

But overall, the manuscript is well written and the methods are thoroughly described such that the paper warrants publication after revisions have been made.

Henning Löwe

**Specific comments**

(p2/l10): In literature there is some inconsistency about *effective*: In the context of upscaling (e.g. Calonne 2011) "effective" is used in the sense of macroscopic, even if conduction only. Sometimes "effective" is used when the mix of conductive and phase-change processes is meant.

(p2/l29): Another discussion of uncertainties can be found in [2].

(p3/l10): Maybe also [1] should be mentioned due to the similarity to the present work. The extension to wet snow might be relevant in the future.

(p3/l6): We have shown in [3] how the structural anisotropy can be quantitatively utilized to correct the bias/scatter in density-based parametrizations for $k$. The parametrization has been confirmed for tundra snowpacks with strong variability in structural anisotropy [4]. In a nutshell, if snow or firn is subject to temperature gradient (TG) metamorphism the structure is reorganized with almost no or little changes in density but with an increase in structural anisotropy that causes e.g. an increase in thermal conductivity. For near surface snow in perennial snowpacks the anisotropy stems from TG metamorphism from the persistent temperature gradient in the top part from the pentration of the annual heat wave into the firn. Anisotropy in near-surface arctic or antarctic firn is well discussed, e.g. in [5]. From this point of view the results obtained in the present paper are consistent with reported influences of structural anisotropy on physical properties in snow/firn.

In principle the optimization method suggested here could be readily utilized to infer the anisotropy parameter $Q$ directly by plugging the parametrization $k(\rho, Q)$ from [3] into the heat equation (3) and subsequently optimizing over $\rho$ and $Q$ instead of $\rho$ and $k$. This would also heal the fact that $k$ and $\rho$ are (erroneously) treated as uncorrelated optimization parameters, while, in contrast, density and anisotropy can be effectively regarded as two independent geometrical features of the microstructure (values for $Q$ should likely lie in the range $[0.33, 0.45]$) It must be kept in mind though that the parametrization from [3] *must* fail somewhere at very high snow/firn densities, but the limit of validity has not been explored yet. While a comprehensive analysis in this direction is certainly beyond the scope of the paper, a simple test in this direction could help to provide some confidence, that the main conclusion is not affectd by existing correlations between $k$ and $\rho$

(p7/l4): Either the functional should be denoted by $F_{\tau,\varrho}$ instead of $F_{\tau,\varsigma}$ or otherwise measured densities denoted by $\varsigma$ (check throughout)

(p7/l14): The statement that there is no data on error estimates of density measurements in snow/firn is a too brave.

(p7/l15): I think at least a quick sensitiviy should be made (i guess that has been done anyway) on the value of $\gamma$. This is directly related to the assumed accurcacy of density measurements.

(p8/l18): $\rightarrow$ ... and $A$ is defined as...

(p13/l6): either replace "reduction" by "increase" or "wrong" by "right"?

(p13/l27): Maybe use $\bar{A}_{ij}$ instead of $\bar{A_{ij}}$, the latter looks like superscript minus...

(p13/l30): Just a thought: The different error pattern is probably an effect of the high variability in the stratigraphy in 2014. From the exactly known *stationary* solution of Eq (3) one can infer that the temperature at a particular location involves the harmonic mean of the entire conductivity profile, which is highly affected by the vertical variability. A similar mathematical structure likely governs the *transient* problem used for the optimization, so I would expect that (synthetically) decreasing the fluctuations in the density profile from 2014 (e.g. using a running mean of measured densities for the optimization) would lead to a different spatio-temporal uncertainty pattern.

(p12): It might be interesting to illustrate the behavior of the cost function, which lives on high-dimensional space, in the vicinity of the minimum, e.g. as a function on a 2d subspace of density and conductivity for one location in space.

(p15/l27): In my opinion, the assumed uncertainty of $1\text{kg/m}^3$ is unrealistically low anyway. (See comment above on $\gamma$) BTW: The method section should maybe include a quick recap on how densities were actually measured from the firn core

**References**

[1] Nicolsky et al, Using in-situ temperature measurements to estimate saturated soil thermal properties by solving a sequence of optimization problems, The Cryosphere, 1, 41-58, 2007

[2] Riche, F., Schneebeli, M. (2010). Microstructural change around a needle probe to measure thermal conductivity of snow. Journal of Glaciology, 56(199), 871-876. doi:10.3189/002214310794457164

[3] Löwe, H., Riche, F., and Schneebeli, M.: A general treatment of snow microstructure exemplified by an improved relation for thermal conductivity, The Cryosphere, 7, 1473-1480, https://doi.org/10.5194/tc-7-1473-2013, 2013.

[4] Gouttevin, I., Langer, M., Löwe, H., Boike, J., Proksch, M., and Schneebeli, M.: Observation and modelling of snow at a polygonal tundra permafrost site: spatial variability and thermal implications, The Cryosphere, 12, 3693-3717, https://doi.org/10.5194/tc-12-3693-2018, 2018.

[5] Fujita, S., Hirabayashi, M., Goto-Azuma, K., Dallmayr, R., Satow, K., Zheng, J., and Dahl-Jensen, D.: Densification of layered firn of the ice sheet at NEEM, Greenland, J. Glaciol., 60, 905921, doi:10.3189/2014JoG14J006, 2014.

---

## Referee Comment (RC2) · Edwin Waddington (Referee) · 5 Mar 2019

[12pt] article geometry a4paper

wasysym

**Journal:** *The Cryosphere Discussions*
**Manuscript:** tc-2018-294: Thermal conductivity of firn at Lomonosovfonna, Svalbard, derived from subsurface temperature measurements
**Authors:** Sergey Marchenko, Gong Chen, Per Lötstedt, Veijo Pohjola, Rickard Pettersson, Ward van Pelt, and Carleen Reijmer

[Figure]

**Reviewer:** Ed Waddington

**1 Overview**

This paper is a thorough and commendable analysis of a set of shallow borehole temperature measurements from Lomonosovfonna, Svalbard, using a least-squares analysis to infer an effective relation between thermal conductivity and density while rigorously accounting for measurement errors in temperature, sensor depth, and firn density. Although the uncertainty analysis is relatively standard, the attention to detail is exceptional.

My major concern is that a"best" relation between conductivity and density may not exist. Although density is likely the most important control on conductivity, I expect that the microstructural texture of the firn is also very important. I would expect that the range of effective conductivities from the various studies (e.g. Figure 7) is due primarily to differing microstructural textures.

For example, firn with large grain-to-grain bonds should conduct heat significanltly better than firn of the same density with less-well-developed bonds. Although this study restricted modeling to periods with no melting of ice or freezing of water, nonetheless the presence of transient meltwater at other times has probably modified the microstructures in significant ways. The Lomonosovfonna conductivity is greater than in Sturm et al. (1997), and in Calonne et al. (2011) at all densities, perhaps due to greater amounts of melting?

Measuring microstructures with micro-CT scans is time consuming and expensive, and is not common (yet), but it may be needed to further advance the conductivity relations. In the meantime, any additional field data (e.g. by hand lens or other tools) on bond size, and grain size or elongation, might provide the next helpful step.

I am not saying this is a fatal flaw for the current paper, only that this point might merit more discussion in the manuscript.

In my view, the paper will be suitable for publication in *The Cryosphere* after revisions.

**1.1 Scientific points**

- Have you considered solving for thermal diffusivity, as colleagues at Dartmouth have done, rather than for conductivity? Since $k$ and $\rho$ are both included in diffusivity, there is only one parameter to find. Does that decrease the scatter in the solutions? In the analysis in the manuscript, $\rho$ seems to show up more as a complication than as a valuable result, and there is no comparison of inferred density against a transient densification model.

  Or is it more important to try to resolve both $k$ and $\rho$?

- The manuscript focuses primarily on comparisons with two other models, i.e. Sturm et al. (1997), and Calonne et al. (2011). However, there are other models in the literature. Below is a brain dump of references relating to conductivity - some of these are already cited, and some are not. Could plotting up predictions from all these other models give readers a better sense of the spread among the current models, and therefore the importance of overlooked physical properties such as microstructure? Perhaps not adding clutter to Figure 7, but making an additional figure?

  Anderson EA (1976) A point energy and mass balance model of a snow cover. (doi:10.1016/S0074-6142(99)80039-4)

  Brandt RE and Warren SG (1997) Temperature measurements and heat transfer in near-surface snow at the South Pole. J. Glaciol. 43(144), 339–351

  Thermal properties and temperature distribution of snow/firn on the Law Dome ice cap, Antarctica. Antarct. Res. 2(2), 38–46

Lüthi MP and Funk M (2001) Modelling heat flow in a cold, high-altitude glacier: Interpretation of measurements from Colle Gnifetti, Swiss Alps. J. Glaciol. 47(157), 314–324 (doi:10.3189/172756501781832223)

Riche F and Schneebeli M (2013) Thermal conductivity of snow measured by three independent methods and anisotropy considerations. Cryosphere 7(1), 217–227 (doi:10.5194/tc-7-217-2013)

Schwander J, Sowers T, Barnola J-M, Blunier T, Fuchs A and Malaizé B (1997) Age scale of the air in the summit ice: Implication for glacial-interglacial temperature change. J. Geophys. Res. Atmos. 102(D16), 19483–19493 (doi:10.1029/97JD01309)

Schwerdtfeger P (1963) Theoretical derivation of the thermal conductivity and diffusivity of snow. IAHS Publ 61, 75–81 http://iahs.info/uploads/dms/061007.pdf

Sturm M, Holmgren J, König M and Morris K (1997) The thermal conductivity of seasonal snow. J. Glaciol. 43(143), 26–41 (doi:10.1017/S0022143000002781)

Van Dusen MS (1929) Thermal conductivity of non-metallic solids. International critical tables of numerical data, physics, chemistry and technology. McGraw-Hill New York, 216–217

Yen Y-C (1981) Review of Thermal Properties of Snow, Ice, and Sea Ice. CRREL Rep. 81-10, 1–27 http://acwc.sdp.sirsi.net/client/search/asset/1005644

- Page 6, Equation (5) -
Equation (5) uses the arithmetic mean of conductivity $k$ at the nodal midpoints. This should be adequate for the exercise here, but for cases where the conductivity gets very small or zero, it is preferable to use the geometric mean

$$\left( \frac{2k_i k_{i+1}}{k_i + k_{i+1}} \right) \tag{1}$$

which actually goes to zero to prevent heat transfer across the interface when one of the bounding conductivites is zero, and prevents heat leakage to a node with very low conductivity. e.g. see page 44 in Patankar (1980), or many other texts.

Patankar, S.V., 1980. *Numerical heat transfer and fluid flow*, Hemisphere.

- Page 7, Equation (6) and line 15 -
The objective function has to be nondimensional, since you are mixing temperature values and density values. Therefore, the weighting term $\sigma_{\rho_i}$ cannot be set to unity as stated - it must be set to some characteristic density factor.

- Page 11, Line 13 -
Why is $L$ assumed to be 1 meter?

- Figures 2 and 4. -
The units on the horizontal axes look impossible. For example, how can Spring 2013 begin only 40 days after Spring 2012 ends, and only 99 days after 21 April, 2012?

**1.2 Editorial points and clarity**

- A table of variables would be helpful for readers.

- *Data* are always plural. The word is often used incorrectly with a singular verb in the text.

- The units of conductivity are usually expressed as W m$^{-1}K^{-1}$. Is there a reason for separating the Watts into Joules per second?

However you decide to do it, at least be consistent. For example, Page 13, line 127 - $J(Kms)^{-1}$ *vs* Page 12, line 24, $J(smK)^{-1}$

- Page 15, line 27 -
  Where does the $K^{-3}$ come from?

- Page 1, Line 13:
  *As a basic physical property of a medium temperature of snow, firn and ice controls multiple processes occurring therein and at the glacier surface.*
  I do not understand what this sentence is trying to say. Perhaps some commas could help?

- Page 2, Line 5:
  *Since most temperature fluctuations occur at the surface, the dominant direction of heat flux is vertical:*
  If temperature fluctuations are due to weather and climate, of course they are maximum at the surface, (particularly when latent-heat effects are negligible), but they actually occur at all depths. What message is this sentence trying to convey?

- Page 12, Line 11 -
  Spell *pattern*

- Page 2, line 4 -
  Cuffey and Paterson is a big book. It helps readers when you include a page number.
  Page 5, line 9 - Same comment.
  Page 6, line 6 - Same comment.
  Page 16, line 14 - Same comment.

- The abbreviation *ca* is used frequently for the latin *circa*, or "approximately". If the authors do not want to say *approximately*, then the appropriate abbreviation is *c.*

- Page 3, Line 20:
  *The accumulation and melt rates estimated respectively from repeated radar surveys (Pälli et al., 2002; Van Pelt et al., 2014) and modeled surface energy and mass fluxes (Van Pelt et al., 2012) are 0.58–0.75 and 0.34 m w.e.year$^{?1}$, respectively.*
  The subject of this sentence is very complicated, such that readers may not recognize it as a complete sentence on first reading. Can you re-write it in a simpler way?

- Page 3, line 20:
  What scale (in meters) is intended by *local-scale variability*?

- Page 9, line 3 -
  Something went wrong with the meters units. $J(smK)^{-1}$?

- Page 15, line 26 -
  Writing kgm$^{-1}$, the gm can look like grams, causing some reader hesitation.

- Page 16, line 12 -
  *Summit* is a named location, so it should be capitalized.

- Figure 4 -
  It would help readers if you could add Spring, Fall, and Year labels, as on Figure 2.

---

## Author Comment (AC1) · 4 Apr 2019

Authors are very grateful to the reviewers for the careful inspection of the original manuscript. The comments gave rise to some thinking, which resulted in corrections and additions both in the text and in the figures of the manuscript. We are also thankful to the tips on usage of symbols, spelling and phrasing utilized in the text. Here below we present answers to comments from both reviewers. The text in blue and green fonts is the feedback on original manuscript kindly provided by Henning Löwe and Edwin Waddington respectively. Replies of the authors to specific comments in given in the black font text found below the comment in question.

**Henning Löwe:**

In the paper the authors retrive profiles of thermal conductivity of near-surface snow/firn in Svalbard as optimization parameters from a comparison of the numerical solution of the heat equation with thermistor (chain) measurements buried over several years. The optimization is restricted to the dry zone.

In my opinion, using intermediate complexity (few parameter) models to constrain physical properties from measurements via optimization is often more conclusive over a mere comparison with more complex (many parameter) models. Even though this method is not new, the work is apparently sound and the authors come to a clear and important conclusion, namely that common density based parametrizations are insufficient for the thermal conductivity in near-surface firn. While I would immediately agree on this funding on physical grounds (see comments), I don't fully understand how the assumption that ρ and k are used as independent optimization parameters (which is certainly not true) may affect this finding. A little more discussion seems required here. This and other things are listed in the comments below.

But overall, the manuscript is well written and the methods are thoroughly described such that the paper warrants publication after revisions have been made.

Specific comments:

(p2/l10): In literature there is some inconsistency about *effective*: In the context of upscaling (e.g. Calonne 2011) *effective* is used in the sense of macroscopic, even if conduction only. Sometimes *effective* is used when the mix of conductive and phase-change processes is meant.

Here we use the term *effective* thermal conductivity following the Sturm et al., 1997 understanding of it. It thus includes both the pure conduction (through the rigid ice matrix and air in pores) and the latent heat fluxes. It is, obviously, not possible to distinguish between the two components using our method. We also think that in the framework of glacier mass balance simulation it is more practical to operate the effective thermal conductivity in its wider sense. The text was rephrased to exclude misinterpretation.

(p2/l29): Another discussion of uncertainties can be found in [2].
True, the text is edited and now contains the reference.

(p3/l10): Maybe also [1] should be mentioned due to the similarity to the present work. The extension to wet snow might be relevant in the future.

True, the text is edited and now contains the reference.

(p3/l6): We have shown in [3] how the structural anisotropy can be quantitatively utilized to correct the bias/scatter in density-based parametrizations for k. The parametrization has been confirmed for tundra snowpacks with strong variability in structural anisotropy [4]. In a nutshell, if snow or firn is subject to temperature gradient (TG) metamorphism the structure is reorganized with almost no or little changes in density but with an increase in structural anisotropy that causes e.g. an increase in thermal conductivity. For near surface snow in perennial snowpacks the anisotropy stems from TG metamorphism from the persistent temperature gradient in the top part from the pentration of the annual heat wave into the firn. Anisotropy in near-surface arctic or antarctic firn is well discussed, e.g. in [5]. From this point of view the results obtained in the present paper are consistent with reported influences of structural anisotropy on physical properties in snow/firn.

In principle the optimization method suggested here could be readily utilized to infer the anisotropy parameter Q directly by plugging the parametrization k($\rho$,Q) from [3] into the heat equation (3) and subsequently optimizing over $\rho$ and Q instead of $\rho$ and k. This would also heal the fact that k and $\rho$ are (erroneously) treated as uncorrelated optimization parameters, while, in contrast, density and anisotropy can be effectively regarded as two independent geometrical features of the microstructure (values for Q should likely lie in the range [0.33, 0.45]) It must be kept in mind though that the parametrization from [3] must fail somewhere at very high snow/firn densities, but the limit of validity has not been explored yet. While a comprehensive analysis in this direction is certainly beyond the scope of the paper, a simple test in this direction could help to provide some confidence, that the main conclusion is not affected by existing correlations between k and $\rho$.

That is a good point. Indeed, the multiannual firn pack at Lomonosovfonna has been subjected to multiple cycles of annual temperature fluctuations and can be expected to have developed structural anisotropy resulting also in a difference in vertical and horizontal thermal conductivity. The short summer season at the study site, yet results in several weeks when the entire firn pack gets temperate (although not necessarily with a direct influence of liquid water on the microstructure), which likely decreased the effect of constructive temperature-gradient metamorphism. Below the depth of ca 1-2 m (roughly the net annual accumulation layer) firn is rather dense and temperature gradient are not as steep as in the near subsurface layers, which altogether does not favor the development of faceted crystals. At the same time another source of structural anisotropy at somewhat larger scales may be seen in the preferential water flow, known to occur at the site. This may be at least one of the explanations for the overall higher optimized k values compared to k=f($\rho$) functions based on measurements in seasonal snow packs. The corresponding part in the discussion is edited.

Referring to the comment above and also to the text in the "General comments" section of the review, we would like to note that the independence between optimized parameters k and $\rho$ is not seen as a problem. The optimization of $\rho$ values is done here only to allow for some flexibility in the density values going into the forward model and by this appreciate the facts of uncertainty in

measured density values and of the "non-representativeness" of single core records at the lateral scales of ca 10 m. The 1D heat conduction equation has three parameters characterizing the media: thermal conductivity, density and specific heat capacity. The first two can vary in a large range and are thus included in the optimization, the latter was not reported to vary much and is thus taken as constant. These considerations were expressed in page 7 line 17-20 of the TC Discussion paper. Secondly, generally speaking, producing realistic values for two independent parameters is a more challenging optimization task compared to a search of two dependent variables. So the fact that the optimization does not «go crazy» and still produces realistic density values is seen as a demonstration of the robustness of the method.

"Optimizing over $\rho$ and $Q$ instead of $\rho$ and $k$", as suggested by the Reviewer is challenging primarily because there is no empirical data to constrain the $Q$ values, like there is for density. The detailed empirical data on snow/firn structure (from e.g. microscope observations, thin sections of 3D microtomography) is not available for the site. $Q$ will be more or less a free parameter.

(p7/l4): Either the functional should be denoted by $F_{T,\varrho}$ instead of $F_{T,\varsigma}$ or otherwise measured densities denoted by $\varsigma$ (check throughout)

That was, indeed, a mistake. The equations and text were edited.

(p7/l14): The statement that there is no data on error estimates of density measurements in snow/firn is a too brave.

Agreed, in a general case it would be, indeed, doubtful to claim that no estimates of accuracy in density measurements are available. What was intended to say is that only one core was drilled every April and therefore we lack empirical data on the basis of which more or less trust can be given to different density measurements along the vertical profiles. The text was edited.

(p7/l15): I think at least a quick sensitiviy should be made (i guess that has been done anyway) on the value of $\gamma$. This is directly related to the assumed accurcacy of density measurements.

Indeed, such tests were done but the results were not included in the original manuscript. A quick sensitivity check by varying $\gamma$ from $10^{-5}$ to $10^{7}$ has shown that too small $\gamma$ values lead to oscillatory optimal $\rho$ values and too large $\gamma$ values tends to give a non-optimal solution. Here below are the results for the 4 spring domains. Results for 2012 are now included in the manuscript along with some insights on behavior of the cost function around its minima,

[Figure]

[Figure]

Agreed, text edited.

(p13/l6): either replace "reduction" by "increase" or "wrong" by "right"?

Agreed, text edited.

(p13/l27): Maybe use $\bar{A}_{ij}$ instead of $\bar{A_{ij}}$, the latter looks like superscript minus...

Agreed, text edited.

(p13/l30): Just a thought: The different error pattern is probably an effect of the high variability in the stratigraphy in 2014. From the exactly known stationary solution of Eq (3) one can infer that the temperature at a particular location involves the harmonic mean of the entire conductivity profile, which is highly affected by the vertical variability. A similar mathematical structure likely governs the transient problem used for the optimization, so I would expect that (synthetically) decreasing the fluctuations in the density profile from 2014 (e.g. using a running mean of measured densities for the optimization) would lead to a different spatio-temporal uncertainty pattern.

Explanation of the spatiotemporal pattern in sensitivity A for the fall 2014 domain has "drunk a lot of blood" from all coauthors. Although the above considerations regarding the vertical links between k and rho values are absolutely valid, the particularly spiky density curve measured in the 2014 core can hardly provide an explanation. The forward model uses the values of k and rho generated at the previous step of the optimization routine (or initial guess in case it is the first iteration of the optimization routine). Thus it is not the spiky blue curve from Fig. 1 but the 1-m spaced smoothed red (yellow for fall 2014) curve that is used in the forward thermal conduction model. The 1-m spaced density values are then piecewise-linearly interpolated to a finer 0.1 spaced grid and the optimization tends to minimize the misfit between this interpolated density profile and the measured profile. This point was not clearly expressed in the original manuscript (see p. 6, l. 27-31), which is now corrected to avoid misinterpretations. It can further be noted that the same density profile is used for the spring 2014 domain, which, however does not show a similar behavior of the sensitivities A.

(p12): It might be interesting to illustrate the behavior of the cost function, which lives on high-dimensional space, in the vicinity of the minimum, e.g. as a function on a 2d subspace of density and conductivity for one location in space.

One way to investigate the cost function close to the optimum is to look at its Hessian matrix H, the matrix of second order derivatives of the 16 variables we use: $\rho_*$, $k_*$. It is composed of the matrices A, B and W in the paper and the eigenvalues of the 16x16 matrix tell about the behavior at the optimum. H is always positive definite and we have found a minimum. If some of the eigenvalues are very small compared to the other ones, then the minimum in the directions

given by the corresponding eigenvectors is not well defined. These results are now included in the updated manuscript as figure 4 and text referring to it.

(p15/l27): In my opinion, the assumed uncertainty of 1 kg/m$^3$ is unrealistically low anyway. (See comment above on) BTW: The method section should maybe include a quick recap on how densities were actually measured from the firn core.

The derived sensitivities are scalable as long as the perturbations are kept relatively small (ca <10% of the variable range). Thus the choice of the value 1 kg m$^{-3}$ for the density perturbation is driven solely by the fact that this is a unit value. If one, for example, is interested in the expected effect of a 5 kg m$^{-3}$ density perturbation, the reported sensitivities can be simply multiplied by 5. That was highlighted at page 8, lines 6-7 and page 15, lines 25-27 of the TC Discussion manuscript.

We see that as an unnecessary redundancy to repeat the details of the field and cold lab routines applied to derive the density measurements used here. The information along with the data from the cores drilled in 2012-2014 was given in the earlier Journal of Glaciology paper (https://doi.org/10.1017/jog.2016.118).

**Edwin Waddington:**

**1 Overview**
This paper is a thorough and commendable analysis of a set of shallow borehole temperature measurements from Lomonosovfonna, Svalbard, using a least-squares analysis to infer an effective relation between thermal conductivity and density while rigorously accounting for measurement errors in temperature, sensor depth, and firn density. Although the uncertainty analysis is relatively standard, the attention to detail is exceptional.

My major concern is that a "best" relation between conductivity and density may not exist. Although density is likely the most important control on conductivity, I expect that the microstructural texture of the firn is also very important. I would expect that the range of effective conductivities from the various studies (e.g. Figure 7) is due primarily to differing microstructural textures.

For example, firn with large grain-to-grain bonds should conduct heat significanltly better than firn of the same density with less-well-developed bonds. Although this study restricted modeling to periods with no melting of ice or freezing of water, nonetheless the presence of transient meltwater at other times has probably modified the microstructures in significant ways. The Lomonosovfonna conductivity is greater than in Sturm et al. (1997), and in Calonne et al. (2011) at all densities, perhaps due to greater amounts of melting?

Measuring microstructures with micro-CT scans is time consuming and expensive, and is not common (yet), but it may be needed to further advance the conductivity relations. In the meantime, any additional field data (e.g. by hand lens or other tools) on bond size, and grain size or elongation, might provide the next helpful step.

I am not saying this is a fatal flaw for the current paper, only that this point might merit more discussion in the manuscript.

We agree that the structure of the snow/firn pack is one of the missing parameters in the k=f(ρ) parameterizations. That was also pointed out by Henning Löwe. However, we do not see how the available empirical data on stratigraphy can be applied in the current study. In the revised manuscript the discussion chapter (4.6) contains more information about the possible relations between the snow/firn structure at the level of grains and ca 0.1-1 m.

In my view, the paper will be suitable for publication in The Cryosphere after revisions.

**1.1 Scientific points**
   • Have you considered solving for thermal diffusivity, as colleagues at Dartmouth have done, rather than for conductivity? Since k and ρ are both included in diffusivity, there is only one parameter to find. Does that decrease the scatter in the solutions? In the analysis in the manuscript, ρ seems to show up more as a complication than as a valuable result, and there is no comparison of inferred density against a transient densification model. Or is it more important to try to resolve both k and ρ?

It appears that the role of density ρ in the optimization routine was not explained thoroughly enough in the original manuscript. With reference to the reply on the comment from Henning Löwe

to page 3, line 6 the following can be repeated: the optimization of ρ values is done here only to allow for some flexibility in the density values going into the forward model and by this appreciate the facts of uncertainty in measured density values and of the "non-representativeness" of single core records at the lateral scales of ca 10 m. The 1D heat conduction equation has three parameters characterizing the media: thermal conductivity, density and specific heat capacity. The first two can vary in a large range and are thus included in the optimization, the latter was not reported to vary much and is thus taken as constant. These considerations were expressed in page 7 line 17-20 of the TC Discussion paper.

There are many other ways to formulate the optimization problem. We chose ρ and k as optimized parameters mainly because both have an empirical proxy to compare with: for ρ it is the density measured in cores and for k it is the measured temperature. We assume no prior knowledge about the relation between the parameters and the linear relation between k and ρ is computed in Fig. 8. Other alternatives to determine ρ and k are
- replace k in (6) by ρκ and solve for the scalar or space dependent diffusivity κ and ρ
- replace ρ in (6) by k/κ and solve for κ and k
- let ρ=ϱ and k= ρκ and insert into (6) and solve for κ
- k depend on ρ and Q as in eq (2) in Löwe et al 2013, replace k in (6) by this formula and optimize for ρ and Q (suggested by H. Löwe)

The optimal solutions will be slightly different in all cases but additional criteria are needed to select the best one.

The optimized k and density values can be used to calculate the thermal diffusivity. These results were reported in the chapter 4.6 of the original manuscript along with the comparisons with the Giese and Hawley (2015).

- The manuscript focuses primarily on comparisons with two other models, i.e. Sturm et al. (1997), and Calonne et al. (2011). However, there are other models in the literature. Below is a brain dump of references relating to conductivity - some of these are already cited, and some are not. Could plotting up predictions from all these other models give readers a better sense of the spread among the current models, and therefore the importance of overlooked physical properties such as microstructure? Perhaps not adding clutter to Figure 7, but making an additional figure?

Anderson EA (1976) A point energy and mass balance model of a snow cover. (doi:10.1016/S0074-6142(99)80039-4)
Brandt RE and Warren SG (1997) Temperature measurements and heat transfer in near-surface snow at the South Pole. J. Glaciol. 43(144), 339–351 Thermal properties and temperature distribution of snow/firn on the Law Dome ice cap, Antarctica. Antarct. Res. 2(2), 38–46
Lüthi MP and Funk M (2001) Modelling heat flow in a cold, high-altitude glacier: Interpretation of measurements from Colle Gnifetti, Swiss Alps. J. Glaciol. 47(157), 314–324 (doi:10.3189/172756501781832223)
Riche F and Schneebeli M (2013) Thermal conductivity of snow measured by three independent methods and anisotropy considerations. Cryosphere 7(1), 217–227 (doi:10.5194/tc-7-217-2013)
Schwander J, Sowers T, Barnola J-M, Blunier T, Fuchs A and Malaizé B (1997) Age scale of the air in the summit ice: Implication for glacial-interglacial temperature change. J. Geophys. Res. Atmos. 102(D16), 19483–19493 (doi:10.1029/97JD01309)

Schwerdtfeger P (1963) Theoretical derivation of the thermal conductivity and diffusivity of snow. IAHS Publ 61, 75–81 http://iahs.info/uploads/dms/061007.pdf

Sturm M, Holmgren J, König M and Morris K (1997) The thermal conductivity of seasonal snow. J. Glaciol. 43(143), 26–41 (doi:10.1017/S0022143000002781)

Van Dusen MS (1929) Thermal conductivity of non-metallic solids. International critical tables of numerical data, physics, chemistry and technology. McGraw-Hill New York, 216–217

Yen Y-C (1981) Review of Thermal Properties of Snow, Ice, and Sea Ice. CRREL Rep. 81-10, 1–27 http://acwc.sdp.sirsi.net/client/search/asset/1005644

The parameterizations k = f(ρ) are numerous and we do not see the review of the past work on the density-thermal conductivity relations as an aim of the present manuscript. The figure below illustrates the spread in the functions found in literature. The revised manuscript contains one additional parameterization: Riche and Schneebeli (2013) based their equation on measurements in faceted and depth hoar crystals and suggest consistently higher k values than the predictions according to Sturm and Calonne k = f(ρ) relations. The updated figure provides more insights on the spread the ρ – k equations in literature.

[Figure]

- Page 6, Equation (5) - Equation (5) uses the arithmetic mean of conductivity k at the nodal midpoints. This should be adequate for the exercise here, but for cases where the conductivity gets very small or zero, it is preferable to use the geometric mean $(2k_i k_{i+1})/(k_i + k_{i+1})$, which actually goes to zero to prevent heat transfer across the interface when one of the bounding conductivites is zero, and prevents heat leakage to a node with very low conductivity. e.g. see page 44 in Patankar (1980), or many other texts.

Patankar, S.V., 1980. Numerical heat transfer and fluid flow, Hemisphere.

True. Yet, we prefer to keep the more standard and straightforward finite difference discretization based on arithmetical mean. It is second order accurate in the grid size and is the best one can achieve involving three grid points. We suggest that the choice is further justified by the fact that the vertical spacing is quite fine (0.1 m), the vertical gradients in k are not large and the minimal optimized k value is 0.2 W/(m K).

- Page 7, Equation (6) and line 15 - The objective function has to be nondimensional, since you are mixing temperature values and density values. Therefore, the weighting term $\sigma_{pi}$ cannot be set to unity as stated - it must be set to some characteristic density factor.

True, in the revised manuscript units of density are added to the weighting term $\sigma_{pi.}$

- Page 11, Line 13 - Why is L assumed to be 1 meter?

The term L is used as a way to help reader imagine what the term $\varepsilon$ is (equation 25 reads: $\varepsilon=L - (L^2-d^2)^{0.5}$), and make the errors $E[\delta k]$ in equation 22 more intuitively understandable. L is set to 1 m to simulate the situation when the thermistor cable bounces from one side of the borehole wall to the other one with a period of 1 m.

- Figures 2 and 4. - The units on the horizontal axes look impossible. For example, how can Spring 2013 begin only 40 days after Spring 2012 ends, and only 99 days after 21 April, 2012?

There is no error in the time scale, but a clarification is indeed needed: the scale is not continuous and time "goes faster" between the domains 1-2, 2-3 and 3-4. To optimize the usage of figure- and page-space, the time gaps between the first 4 domains are set to 40 days. To exclude misinterpretation of the figure, corresponding remarks are added to captions of figures 2 and 4.

1.2 Editorial points and clarity
- A table of variables would be helpful for readers.

Care was taken make sure that all symbols appearing in equations come with a proper explanation to avoid misintepretations. We suggest that it would be excessive to have a table of variables in this manuscript. However, in case the The Cryosphere Journal allows to have tables listing used symbols/variables in the published papers and the Reviewer repeatedly encourages authors to include such a table, we will be happy to provide it for the convenience of the potential readers.

- Data are always plural. The word is often used incorrectly with a singular verb in the text.

Agreed, corresponding editions are introduced.

- The units of conductivity are usually expressed as W (m K)$^{-1}$. Is there a reason for separating the Watts into Joules per second? However you decide to do it, at least be consistent. For example, Page 13, line 127 - J(Kms)$^{-1}$ vs Page 12, line 24, J(smK)$^{-1}$.

There was no specific reason for using [J (s m K)$^{-1}$] apart from it being the most basic representation of the parameter units. In the revised manuscript we consistently use [W] in place of [J/s].

- Page 15, line 27 - Where does the K$^{-3}$ come from?

Agreed, that was a mistake originating from copy-pasting the units in Latex, corresponding editions are introduced.

- Page 1, Line 13: As a basic physical property of a medium temperature of snow, firn and ice controls multiple processes occurring therein and at the glacier surface. I do not understand what this sentence is trying to say. Perhaps some commas could help?

Agreed, comma added after "medium".

- Page 2, Line 5: Since most temperature fluctuations occur at the surface, the dominant direction of heat flux is vertical: If temperature fluctuations are due to weather and climate, of course they are maximum at the surface, (particularly when latent-heat effects are negligible), but they actually occur at all depths. What message is this sentence trying to convey?

Agreed, the aim with the phrase was to justify usage of 1D heat conduction equation in place of a 3D equation which applies in a general case and is given earlier. A phrase clarifying that the point of the sentence is to state that the heat flux in near surface layers of a glacier is meant.

- Page 12, Line 11 - Spell pattern

Agreed, text edited.

- Page 2, line 4 - Cuffey and Paterson is a big book. It helps readers when you include a page number. Page 5, line 9 - Same comment. Page 6, line 6 - Same comment. Page 16, line 14 - Same comment.

Agreed, text edited.

- The abbreviation ca is used frequently for the latin circa, or "approximately". If the authors do not want to say approximately, then the appropriate abbreviation is c.

Yes, everywhere, where "*ca*" is used, "approximately" is meant.

- Page 3, Line 20: The accumulation and melt rates estimated respectively from repeated radar surveys (Pälli et al., 2002; Van Pelt et al., 2014) and modeled surface energy and mass fluxes (Van Pelt et al., 2012) are 0.58–0.75 and 0.34 m w.e.year$^{-1}$, respectively. The

subject of this sentence is very complicated, such that readers may not recognize it as a complete sentence on first reading. Can you re-write it in a simpler way?

Agreed, text edited to present the information in two separate sentences.

• Page 3, line 20: What scale (in meters) is intended by local-scale variability?

The lateral scales of variability in stratigraphy studied in by Marchenko et al. (2017a) is 10 m. Text is edited.

• Page 9, line 3 - Something went wrong with the meters units. $J(smK)^{-1}$?

Exactly, there were some issues with Latex commands. Text is edited.

• Page 15, line 26 - Writing $kgm^{-1}$, the gm can look like grams, causing some reader hesitation.

Yes, there were some spaces missing. Text is edited to exclude misinterpretation.

• Page 16, line 12 - Summit is a named location, so it should be capitalized.

Agreed, text edited.

• Figure 4 - It would help readers if you could add Spring, Fall, and Year labels, as on Figure 2.

We now added a remark to the figure caption, stating that the horizontal axis is the same as in Figure 2. We think that adding multiple text labels to the figure will make figure too busy, it is already packed quite densely.